# Near-infrared light and tumor microenvironment dual responsive size-switchable nanocapsules for multimodal tumor theranostics

Zhiyi Wang[1,2,6], Yanmin Ju[3,6], Zeeshan Ali [1], Hui Yin[4], Fugeng Sheng[4]*, Jian Lin[5], Baodui Wang[2]* & Yanglong Hou [1]*

Smart drug delivery systems (SDDSs) for cancer treatment are of considerable interest in the field of theranostics. However, developing SDDSs with early diagnostic capability, enhanced drug delivery and efficient biodegradability still remains a scientific challenge. Herein, we report near-infrared light and tumor microenvironment (TME), dual responsive as well as size-switchable nanocapsules. These nanocapsules are made of a PLGA-polymer matrix coated with Fe/FeO core-shell nanocrystals and co-loaded with chemotherapy drug and photothermal agent. Smartly engineered nanocapsules can not only shrink and decompose into small-sized nanodrugs upon drug release but also can regulate the TME to overproduce reactive oxygen species for enhanced synergistic therapy in tumors. In vivo experiments demonstrate that these nanocapsules can target to tumor sites through fluorescence/magnetic resonance imaging and offer remarkable therapeutic results. Our synthetic strategy provides a platform for next generation smart nanocapsules with enhanced permeability and retention effect, multimodal anticancer theranostics, and biodegradability.

[1] Beijing Key Laboratory for Magnetoelectric Materials and Devices, Department of Materials Science and Engineering, College of Engineering, Beijing Innovation Centre for Engineering Science and Advanced Technology, Peking University, 100871 Beijing, China. [2] State Key Laboratory of Applied Organic Chemistry, Key Laboratory of Nonferrous Metal Chemistry and Resources Utilization of Gansu Province, Lanzhou University, Gansu 730000 Lanzhou, China. [3] College of Life Science, Peking University, 100871 Beijing, China. [4] Department of Radiology, the Fifth Medical Centre, Chinese PLA General Hospital, 100071 Beijing, China. [5] Synthetic and Functional Biomolecules Center, Department of Chemical Biology, College of Chemistry and Molecular Engineering, Peking University, 100871 Beijing, China. [6] These authors contributed equally: Zhiyi Wang, Yanmin Ju. *email: fugeng_sheng@163.com; wangbd@lzu.edu.cn; hou@pku.edu.cn

Smart drug delivery systems (SDDSs) have emerged as promising tools for the treatment of malignancies[1–4]. The key features of ideal SDDSs are the enhanced permeability and retention effect (EPR effect) across complex biological systems, "on-demand" drug release capability, and excellent biocompatibility and biodegradatibility[5–12]. As far as nanomedicine is concerned, on-demand change of size of SDDSs in different environments is critical for efficient transportation of nanocarriers to tumor location[7,13]. Generally speaking, nanoparticles (NPs) with a size of 100–200 nm can improve the circulatory half-life, but they are not easy to penetrate deep cellular layers near the tumor vessels[1,14,15]. On the contrary, small size NPs with a diameter of 4–20 nm easily penetrate into deep tumor tissues, but they are more prone to rapid clearance and insufficient drug retention. To tackle these biological barriers, nanocapsules with size switchable function in different biological environments have been developed. With a large initial size of these nanocapsules can achieve effective EPR-derived tumor accumulation during blood circulation, and then these nanocapsules responsively shrink and become small sized once inside the tumor[7,13,16]. However, to the best of our knowledge, reports on such SDDSs are very limited.

Meanwhile, an emerging area of cancer treatment is designing a series of drugs, which can utilize tumor microenvironment (TME) to improve the effect of tumor therapy[17–19]. It is well known that the rapid growth of tumor cells and distortion of tumor blood vessels often results in insufficient oxygen supply and acidification in solid tumors. Hypoxia and acidic pH in the TME not only accelerate angiogenesis and metastasis of tumors, but also lead to therapeutic resistance and ineffective tumor treatment[20–23]. In particular with regard to photodynamic therapy (PDT) and chemodynamic therapy (CDT), in which oxygen participates in cell killing process, the presence of hypoxia in TME will cause the failure of cancer treatment[24–27]. Recently, various innovative approaches have been explored to overcome the therapeutic resistance caused by hypoxia. Such as, using oxygen "shuttles" (e.g., perfluorocarbon) to deliver oxygen into tumors, or in situ oxygen generation inside the tumor with catalysts, which show promising results in improving the efficacy of PDT and CDT[8–11,24]. However, there are still many challenges in developing TME-sensitive SDDSs to overcome the anaerobic problem and improve the therapeutic effect.

Moreover, according to the requirements of US Food and Drug Administration (FDA), theranostic agents should be completely cleared from the body within a reasonable period of time. Poly (lactic-co-glycolic) acid (PLGA) is the typical polymer approved by US FDA, which offers several advantages in the design of SDDSs. It has been reported that the nanocarriers formed by thermosensitive PLGA can change shape and release payloads quickly when the system temperature is higher than their phase-transition temperature[28]. Simultaneously, as a clinical diagnostic and near-infrared (NIR) fluorescence imaging agent, indocyanine green (ICG) has been approved by the US FDA. ICG has been studied extensively in PDT and photothermal therapy (PTT) due to its remarkable NIR optical properties in the best light transmission window for biomedical applications[29–34]. Iron-based magnetic nanocrystals (NCs) with magnetic resonance imaging (MRI) are known to have high catalytic activity toward $H_2O_2$ to generate oxidative hydroxyl radicals (•OH) and would decompose under acidic pH, showing great potential applications in CDT[26]. We hypothesize that the combination of these materials into "all in one" SDDSs may provide for enhanced tumor accumulation and highly effective diagnostics and therapeutics.

Herein, PLGA–polyethylene glycol–poly (N-isopropyl acrylamide), termed as PPP, was synthesized and tethered with Fe/FeO NCs to form Fe/FeO–PPP heterostructures. Doxorubicin (DOX) and ICG were co-loaded into Fe/FeO–PPP heterostructures to develop flexible DOX–ICG@Fe/FeO–PPP nanocapsules by water–oil–water (W/O/W) emulsion method. The resulting nanocapsules, on the one hand, can in situ overproduce reactive oxygen species (ROS) by reacting with endogenous $H_2O_2$ in tumors, which is expected to overcome the tumor hypoxia-related drug resistance of PDT and chemotherapy. On the other hand, DOX–ICG@Fe/FeO–PPP nanocapsules could shrink and decompose into small-sized nanodrugs triggered by photothermal effect of ICG under laser irradiation and lower pH value in TME. These capabilities show significantly enhanced intratumoral permeability to further improve the therapeutic effect in combination therapy with chemodynamic, photodynamic and chemotherapy. Owing to Fe/FeO NCs as MRI contrast agent and ICG as NIR imaging agent, DOX–ICG@Fe/FeO–PPP–FA nanocapsules can achieve imaging-guided synergetic therapy, which can provide essential information, including tumor size and location, optimal treatment time window and real-time efficacy evaluation.

## Results

**Characterization of DOX–ICG@Fe/FeO–PPP nanocapsules.** The procedure for the synthesis of DOX–ICG@Fe/FeO–PPP nanocapsules is presented in Fig. 1a. Firstly, monodispersed Fe/FeO NCs were synthesized by seed-mediated growth method with thermal decomposition in oil phase. A temperature-responsive multiblock polymer PLGA–PEG–PNIPAM (PPP) was synthesized by covalent bonding subsequently (Fig. 1b and Supplementary Fig. 1). The formation of PPP was confirmed by the Fourier transform infrared (FTIR) spectrometer, which is shown in Supplementary Fig. S2. The red shift of the absorption peak for the stretching vibration of the C = O from carboxyl group (1635 cm$^{-1}$) to amide bond (1689 cm$^{-1}$) proves the amination of folic acid (FA) molecule (Supplementary Fig. 2a: i and ii). The existence of vibration absorption peaks (3410 cm$^{-1}$ and 1480 cm$^{-1}$) for N–H bond (Supplementary Fig. 2a: iv) proved the successful synthesis of PLGA–FA. The formation of amide in PPP was also proved in Supplementary Fig. 2b. Soon afterwards Fe/FeO NCs were tethered with PPP to form Fe/FeO–PPP heterostructures. Then, DOX and ICG were co-loaded into Fe/FeO–PPP heterostructures to develop flexible DOX–ICG@Fe/FeO–PPP nanocapsules by W/O/W emulsion method, which were used for subsequent experiments[35]. Water–oil ratio ($V_{dichloromethane}/V_{water}$) for the initial emulsion and PVA are two key conditions in the synthesis of nanocapsules. Water–oil ratio for the initial emulsion could affect the size and hollowness of the nanocapsule (Supplementary Fig. 3a–e), and $V_{dichloromethane}/V_{water} = 1:4$ was selected in the following experiments. The presence of PVA could affect the dispersion of nanocapsules (Supplementary Fig. 3f). With the increase of the amount of Fe/FeO NCs, the number of NCs on the shell also increased (Supplementary Fig. 4). Finally, we chose to add 3 mg of Fe/FeO NCs (Supplementary Fig. 4b).

The core–shell structure of monodispersed Fe/FeO NCs is clearly revealed in transmission electron microscope (TEM) images (Fig. 2a). The Fe/FeO NCs conisted of a core of Fe NPs (of ~8 nm diameter) with a shell of ~5 nm thick FeO as indicated by high-resolution TEM (HRTEM) in Fig. 2b. TEM images of DOX–ICG@PPP and DOX–ICG@Fe/FeO–PPP nanocapsules are presented in Fig. 2c, d, respectively. The average sizes of DOX–ICG@PPP nanocapsules are $203.8 \pm 45.7$ nm (Supplementary Fig. 5a) while the DOX–ICG@Fe/FeO–PPP nanocapsules are $218.9 \pm 52.1$ nm (Supplementary Fig. 5b). Furthermore, clear lattice fringes of 0.183 nm (core) and 0.249 nm (shell) can be ascribed to the (200) plane of Fe and (111) plane of FeO, respectively, which are consistent with the X-ray diffraction (XRD) results (Fig. 2e). Element mapping analysis further

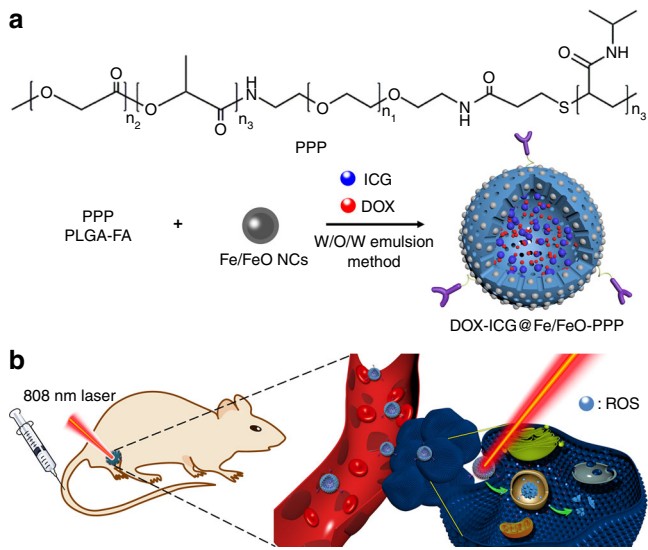

**Fig. 1** Synthesis and biomedical application of DOX–ICG@Fe/FeO–PPP nanocapsules. **a** Fabrication of DOX–ICG@Fe/FeO–PPP nanocapsules. **b** Schematic representation of nanocapsules for imaging-guided stimuli-responsive chemo/photo- and chemodynamic therapy. PPP Poly (lactic-co-glycolic) acid–polyethylene glycol–poly (N-isopropyl acrylamide)

confirmed the core–shell structure of monodispersed Fe/FeO NCs (Fig. 2b). The presence of satellite peaks in the high-resolution X-ray photoelectron spectroscopy (XPS) of Fe 2p shown in Fig. 2f demonstrated the existence of $Fe^{2+}$, supporting the existence of FeO phase in the NCs[36,37]. In addition, XPS of Fe/FeO NCs and DOX–ICG@Fe/FeO–PPP nanocapsules are provided in Supplementary Fig. 6 and Supplementary Fig. 7, which could be determined no significant change in the Fe/FeO NCs after compounding with PPP. The values of saturated magnetic intensity for Fe/FeO NCs and DOX–ICG@Fe/FeO–PPP nanocapsules were 74.4 emu g$^{-1}$ and 14.9 emu g$^{-1}$ in Supplementary Fig. 8a. Therefore, DOX–ICG@Fe/FeO–PPP nanocapsules are highly expected to be $T_2$-weighted MRI contrast agents. In addition, the absorption peaks in the NIR region are shown in the UV-vis absorption spectra of the DOX–ICG@Fe/FeO–PPP nanocapsules (Supplementary Fig. 8b), which means the nanocapsules can be the PTT agents (Fig. 1b).

**Evaluation and mechanism of the ROS generation.** The tumor could be finally destroyed by induced death, which is associated with increased levels of intracellular ROS[38–43]. $^1O_2$ and •OH are two important components in ROS. Therefore, the ability of ROS generation of biomedical materials can be estimated through the quantitative detection of the concentrations of •OH and $^1O_2$. The $^1O_2$ generation amount of ICG@Fe/FeO–PPP nanocapsules in vitro was evaluated by 1,3-diphenylisobenzofuran (DPBF) with pH values of 7.4 (Supplementary Fig. 9a), 6.5 (Supplementary Fig. 9b) and 5.4 (Supplementary Fig. 9c)[44]. $^1O_2$-generation capability of the nanocapsules at a pH value of 5.4 was higher than that at 6.5 and 7.4, because the calculated rate constants were $1.29 \times 10^{-3}$ s$^{-1}$ (pH = 5.4), $3.65 \times 10^{-4}$ (pH = 6.5) and $4.34 \times 10^{-5}$ s$^{-1}$ (pH = 7.4) (Fig. 3a). Iron-based nanomaterials are reported to start the following reactions (Eqs. 1–3) in TME[40–42]:

$$Fe^{2+} + H_2O_2 \rightarrow Fe^{3+} + •OH + OH^- \tag{1}$$

$$Fe + 2Fe^{3+} \rightarrow 3Fe^{2+} \tag{2}$$

$$Fe^{3+} + H_2O_2 \rightarrow Fe^{2+} + •OOH + H^+ \tag{3}$$

In order to quantitatively evaluate the •OH production level of ICG@Fe/FeO–PPP nanocapsules, terephthalic acid (TA) oxidation was selected to measure it by fluorescence spectrum (Supplementary Fig. 10 and Supplementary Fig. 11)[45]. It was divided into four groups: (1) TA + Laser; (2) ICG@PPP + TA + Laser; (3) Fe/FeO + TA + Laser; (4) ICG@Fe/FeO-PPP + TA + Laser. These results showed that ICG@Fe/FeO–PPP + TA + Laser group produced maximum fluorescence enhancement at the same time, which confirmed ICG@Fe/FeO–PPP exhibited the strongest ability to produce •OH in these four groups (Fig. 3b). In addition, the standard consumption of $H_2O_2$ was used to measure the amount of •OH produced by a spectrophotometric method using copper(II) ion and 2,9-dimethyl-1,10-phenanthroline (DMP) (Supplementary Fig. 12)[46]. UV-vis absorption results in Supplementary Fig. 13 demonstrated the synergistic effect of •OH production in ICG@Fe/FeO–PPP nanocapsules, comparing with ICG@PPP nanocapsules and Fe/FeO NCs under the irradiation of 808 nm laser. Electron spin resonance (ESR) spectroscopy further confirmed the potential of the ICG@Fe/FeO–PPP nanocapsules to act as a trigger for the generation of •OH with 5, 5-dimethyl-1-pyrroline N-oxide (DMPO) as a spin trap (Fig. 3c). The increased ESR signal intensity at lower pH values indicates the generation of a great number of •OH, because the Fe/FeO shell of the ICG@Fe/FeO–PPP nanocapsules could act as efficient Fenton catalyst in acidic environment[47]. When incubating free ICG and ICG@Fe/FeO–PPP–FA nanocapsules with oral squamous carcinoma KB cell under the normoxic condition, a similar amount of ROS was generated 1 min after the irradiation of 808 nm laser. More interestingly, the ROS level of cells in the hypoxic condition treated with ICG@Fe/FeO–PPP–FA nanocapsules changed only a little in comparison with those under normoxic condition, while cells treated with free ICG showed much lower ROS levels than those under normoxic condition (Fig. 3d). Furthermore, $^1O_2$ generation was also evaluated by flow cytometry (Supplementary Figs. 14–16). These results of flow cytometry were consistent with those of confocal fluorescence imaging result in Fig. 3d. Additionally, we proposed an explanation of the ROS production of ICG@Fe/FeO–PPP–FA nanocapsules in tumor cells from two aspects (Fig. 3e). On the one hand, Fe/FeO NCs on the surface of the nanocapsules produced large amounts of •OH by catalyzation of $H_2O_2$ in TME. On the other hand, the ICG inside the nanocapsules generated a certain amount of $^1O_2$ by PDT process under the irradiation of 808 nm laser. The two synergistic factors ensure that the nanocapsules could produce ROS even in the hypoxia area of tumor through interaction.

**DOX delivery evaluation of DOX–ICG@Fe/FeO–PPP nanocapsules.** DOX was loaded into the nanocapsules by W/O/W emulsion method. The amount of loaded DOX was quantified by UV-vis absorption spectrum, which could show a distinguished absorption peak appearing at the wavelength of 481 nm (Supplementary Fig. 17). The saturated loading capacity of DOX–ICG@Fe/FeO–PPP nanocapsules is 18.3% by calculation (Supplementary Fig. 18), illustrating that ICG@Fe/FeO–PPP nanocapsules are good candidates as drug carriers.

The photothermal efficiency of ICG@Fe/FeO–PPP nanocapsules was investigated under 808 nm laser irradiation based on the UV-vis absorbance spectra of the nanocapsules (Supplementary Fig. 19a). For an optimal concentration (of just 60 mg L$^{-1}$) of nanocapcules, the laser power of 0.3 W cm$^{-2}$ was employed to obtain a most suitable treatment temperature of 51.5 °C (Supplementary Fig. 19b and c). Interestingly, the ICG@Fe/FeO–PPP nanocapsules were able to maintain good photothermal effects even after several irradiation cycles (Supplementary Fig. 19d). During the laser irradiation process, the diameters of

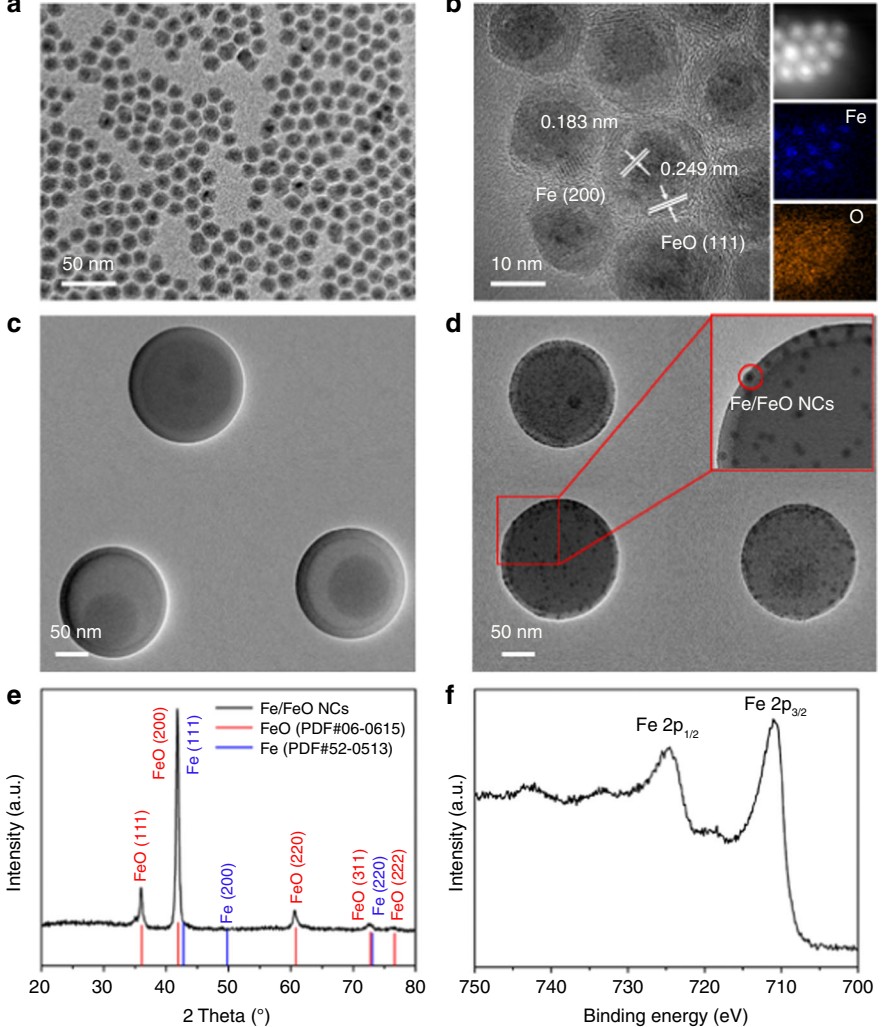

**Fig. 2** Characterization of DOX–ICG@Fe/FeO–PPP nanocapsules. **a** TEM image of Fe/FeO NCs. **b** HRTEM image of Fe/FeO NCs, inset: EDS-mapping of Fe/FeO NCs. **c** TEM image of DOX–ICG@PPP nanocapsules, **d** TEM image of DOX-‐ICG@Fe/FeO–PPP nanocapsules. Inset: HRTEM image of DOX–ICG@Fe/FeO–PPP nanocapsules. **e** XRD patterns of Fe/FeO NCs. **f** High-resolution XPS of Fe 2p for Fe/FeO NCs

nanocapsules were contracted sustainably, which enabled the nanocapsules to release DOX in large amounts due to high temperature stimuli (Supplementary Figs. 19e, f and 20). These results proved that the nanocapsules successfully produced a significant temperature increase under laser irradiation.

To evaluate the release efficiency of DOX–ICG@Fe/FeO–PPP nanocapsules, these nanocapsules were incubated in a phosphate buffer with pH 7.4, 6.5 or 5.4 under the 808 nm laser irradiation. In Fig. 4a, DOX-ICG@PPP nanocapsules did not show obvious difference in drug release at pH 7.4, 6.5 or 5.4. However, DOX–ICG@Fe/FeO-PPP nanocapsules released up to 79.0% at pH 5.4 after 80-h treatment, while the release was limited to 67.4% at pH 6.5 and 55.4% at pH 7.4. The increase of drug release of DOX–ICG@Fe/FeO–PPP nanocapsules is associated to the instability of Fe/FeO NCs in weak acidic condition. Meanwhile, DOX–ICG@Fe/FeO–PPP nanocapsules and DOX-PPP@PPP nanocapsules were exposed to laser irradiation for 5 min (laser on) in a buffer with pH 7.4, 6.5 or 5.4 and then incubated for 55 min (Fig. 4b). DOX–ICG@PPP nanocapsules did not respond to the laser irradiation and pH stimuli, which corresponded with the above results (pH = 7.4: 20.4%, pH = 6.5: 23.4%, pH = 5.4: 26.0%), while abrupt release from DOX-ICG@Fe/FeO–PPP nanocapsules was detected at

pH 7.4 (42.2%), 6.5 (58.2%) and 5.4 (73.9%) when NIR irradiation was repeated after every 1 h.

**Dual-stimulus responsive ICG@Fe/FeO–PPP nanocapsules.** The TEM images of ICG@Fe/FeO–PPP nanocapsules show obvious shrinkage from $220.9 \pm 25.5$ nm to $161.9 \pm 21.8$ nm at pH 6.5 in 24 h upon laser irradiation, which demonstrates the effect of irradiation on the morphology of nanocapsules (Fig. 4c and Supplementary Fig. 21). After 48 h, small size nanocapsules ($54.5 \pm 4.1$ nm) were observed as presented in the TEM images in Fig. 4c. Based on the above results, we deduced that the radial stress analysis of flexible DOX-ICG@Fe/FeO–PPP nanocapsules causes the shrinking process (Fig. 4d). Subsequently, the change of ICG@Fe/FeO–PPP nanocapsules within 1 week after laser irradiation were measured by TEM at pH 6.5 (Supplementary Fig. 22). Moreover, bio-TEM images of KB cells incubated with ICG@Fe/FeO–PPP nanocapsules was also provided in Supplementary Fig. 23. These results proved the instability of ICG@Fe/FeO–PPP nanocapsules after laser irradiation in weak acidic condition. We hypothesized that the stability of this system was caused by large-scale local buckling instability of the nanocapsules after the irradiation of the laser[48]. The irradiation of the

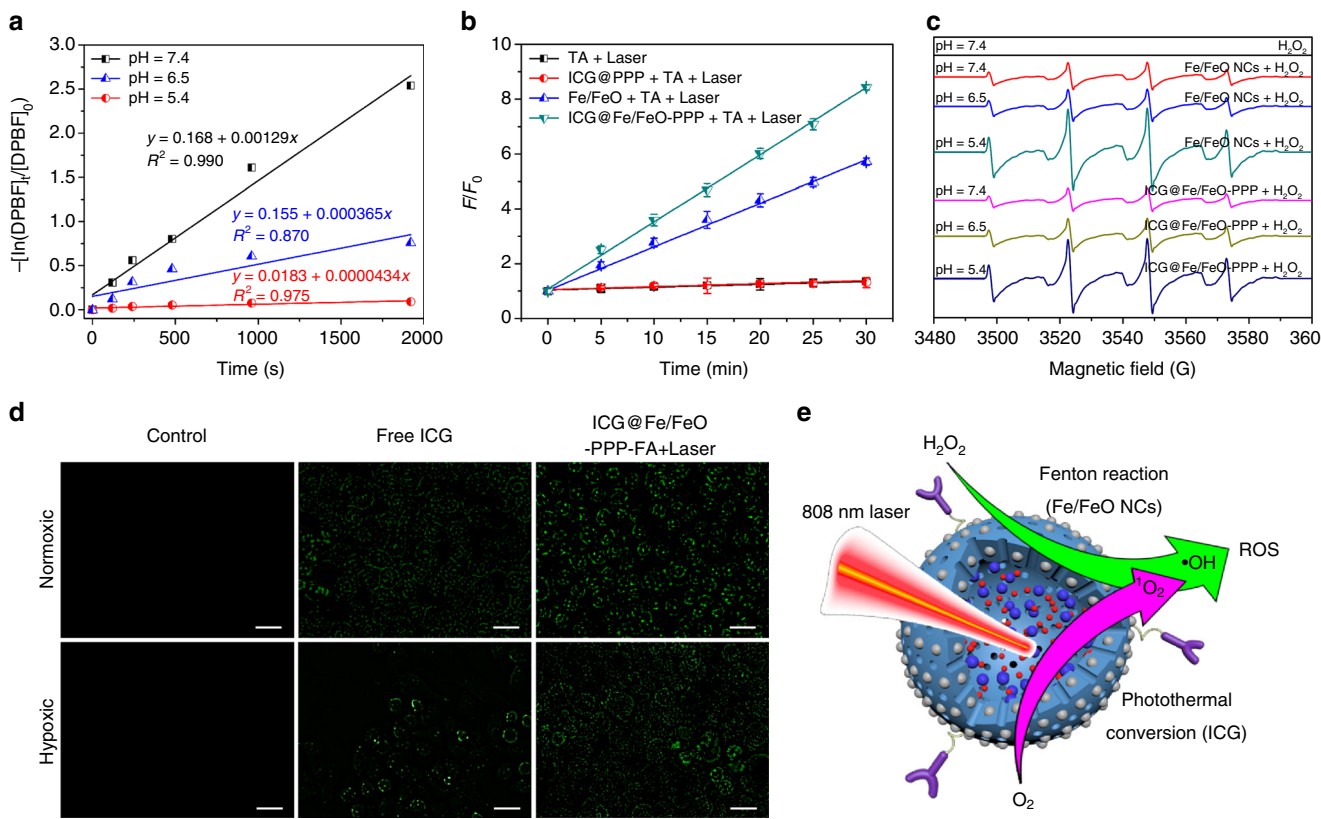

**Fig. 3** Evaluation of the ROS generation and the mechanism of ROS production. **a** Detection of $^1O_2$ by UV-vis absorbance spectra for ICG@Fe/FeO-PPP nanocapsules with DPBF at pH value of 7.4, 6.5 and 5.4 phosphate buffer. **b** Fluorescence intensity of TAOH at 440 nm as a function of laser irradiation time for the ICG@Fe/FeO-PPP nanocapsules and different control treatments. $F_O$ and $F$ were the fluorescence intensities of the system without or with treatment, respectively. The error bars represent the standard deviations of three separate measurements. **c** ESR spectra of Fe/FeO NCs and ICG@Fe/FeO-PPP nanocapsules with DMPO as the spin trap in different pH values condition (7.4, 6.5 and 5.4). **d** $^1O_2$ generation evaluated by DHR123 in KB cells treated with free ICG and ICG@Fe/FeO-PPP-FA nanocapsules under normoxic and hypoxic conditions (scale bars: 20 μm). **e** Synergism schematic of Fenton reaction of Fe/FeO NCs with photothermal conversion (ICG) in ICG@Fe/FeO-PPP-FA nanocapsules. (Error bars, mean ± SD, $n = 6$, ROS reactive oxygen species.)

laser could lead to the temperature increase inside of the nanocapsule, which further caused the change in interaction between the internal interface of nanocapsule and water molecule by intermolecular forces of PPP[49,50]. Due to the degradability of ester bond and amide bond in PPP at pH below 6.5 (Fig. 4e), the shrinking nanocapsules were eventually hydrolyzed[51–56].

**In vivo imaging and biodistribution**. The cytotoxicity and photothermal ability of ICG@Fe/FeO-PPP nanocapsules at cellular level were evaluated by using NIH3T3 and KB cells. Due to the overexpression of the folate receptor in KB cells, ICG@Fe/FeO-PPP-FA nanocapsules can specifically recognize the KB cells through interactions between FA and folate receptor (on the surface of cells). Supplementary Fig. 24a shows that the cell viability of NIH3T3 and KB cells, incubated with the ICG@Fe/FeO-PPP-FA nanocapsules (of Fe concentrations up to 3.20 mM), could reach even higher than 95%, which is clear indication of very low cytotoxicity of nanocapsules. Subsequently, we examined the photothermal efficiency of each nanocapsules in KB cells. As a result, a more significant laser-induced photothermal killing efficiency of KB cells was observed in ICG@Fe/FeO-PPP-FA nanocapsules with the irradiation of laser (Supplementary Fig. 24b). Furthermore, live/dead cell staining experiments were used to evaluate the laser-triggered photothermy effect. As shown in Supplementary Fig. 24c, red fluorescence appeared only for the cells treated with DOX–ICG@Fe/FeO–PPP–FA nanocapsules

under the irradiation with laser, which is consistent with the CCK8 assay results in Supplementary Fig. 24b. These results demonstrated that DOX–ICG@Fe/FeO–PPP–FA nanocapsules can specifically kill the KB cells by PTT.

Iron NPs have the potential to be the agents for $T_2$-weighted MRI. The $r_2$ value of ICG@Fe/FeO–PPP nanocapsules was around 130.7 mM$^{-1}$ s$^{-1}$ when dispersed in water and decreased to 75.9 mM$^{-1}$ s$^{-1}$ when incubated with KB cells (Supplementary Fig. 25). We also assessed the $T_2$-weighted MRI capability in vivo after intravenous injection of ICG@Fe/FeO-PPP and ICG@Fe/FeO-PPP-FA nanocapsules (20 mg kg$^{-1}$, 200 mL) into KB tumor-bearing nude mice. Figure 5a, b clearly indicates that the ICG@Fe/FeO–PPP–FA nanocapsules show more stronger signal intensity and make the tumor darker than ICG@Fe/FeO–PPP after 24 h of injection. These results suggest higher accumulations of ICG@Fe/FeO–PPP–FA at the tumor sites owing to the active targeting.

Furthermore, fluorescence imaging was carried out to track the in vivo behaviors of ICG@Fe/FeO–PPP and ICG@Fe/FeO-PPP-FA nanocapsules (20 mg kg$^{-1}$, 200 mL) after intravenous injection into tumor model. ICG could act as fluorescent molecule with the excitation wavelength of 780 nm and emission wavelength of 810 nm (Supplementary Fig. 26). Strong intensity of fluorescence signals were mainly showed for ICG@Fe/FeO-PPP-FA nanocapsules in the tumor site after 12 h (Fig. 5c). In contrast, no obvious fluorescence signal appeared in the tumor site for ICG@Fe/FeO–PPP nanocapsules even after 24 h.

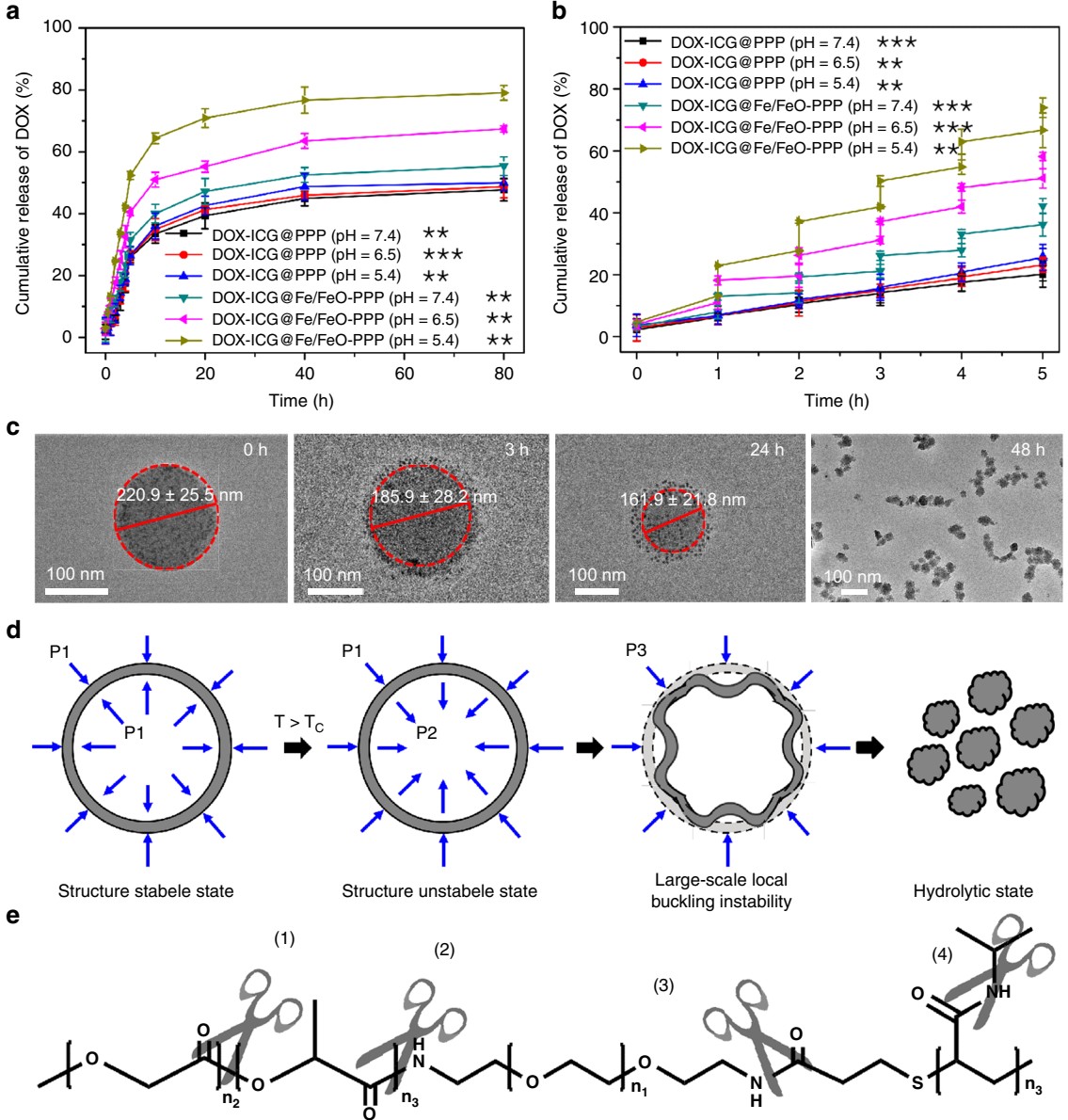

**Fig. 4** DOX delivery evaluation of DOX–ICG@Fe/FeO-PPP nanocapsules. **a** DOX release curves of DOX-ICG@PPP nanocapsules and DOX-ICG@Fe/FeO-PPP nanocapsules in different pH values (7.4, 6.5 and 5.4) at 37 °C. **b** DOX release curves of DOX-ICG@PPP nanocapsules (pH = 7.4, 6.5 and 5.4) and DOX-ICG@Fe/FeO-PPP nanocapsules (pH = 7.4, 6.5 and 5.4) with four laser on/off cycles under a laser irradiation (808 nm, 0.3 W cm$^{-2}$) for 5 min. **c** TEM image of the shrinking process for DOX-ICG@Fe/FeO-PPP nanocapsules in 48 h after the irradiation of laser (808 nm, 0.3 W cm$^{-2}$) for 5 min (pH = 6.5). **d** Schematic diagram of radial stress analysis of DOX-ICG@Fe/FeO-PPP nanocapsules in the shrinking process, and P1, P2 and P3 represent the radial stress for the nanocapsules in different stages. **e** Schematic diagram of the degradability for PPP at pH below 6.5. *P* values in **a** and **b** were calculated by Tukey's post-hoc test (\*\**P* < 0.01, \*\*\**P* < 0.001) by comparing other groups with the last group (DOX-ICG@Fe/FeO-PPP-FA, pH = 5.4). Error bars, mean ± SD (*n* = 6)

Moreover, the targeting capacity of ICG@Fe/FeO–PPP nanocapsules and ICG@Fe/FeO–PPP–FA nanocapsules was evaluated by ex vivo imaging of main organs (liver, spleen, lung, heart and kidney) and tumors of mice after 48 h of intravenous injection. Fluorescence signals were clearly observed in the tumor and liver also have strong signals; however, spleen, lung, heart and kidney show no obvious fluorescence signals (Fig. 5c, d). These results suggested that ICG@Fe/FeO–PPP–FA nanocapsules accumulated much more than ICG@Fe/FeO–PPP nanocapsules at the tumor sites. In addition, active targeting based on FA and passive targeting from EPR effect, simultaneously played an important role in this nanomaterials system[57–59]. Moreover, due to the spatial resolution of MRI, we tested real-time MRI of KB tumor-

bearing mice without and without laser irradiation after intravenous injection of DOX–ICG@Fe/FeO–PPP–FA nanocapsules (Supplementary Fig. 27). These results confirm that laser-trigged shrinkage of DOX–ICG@Fe/FeO–PPP nanocapsules was helpful for the deep tumor tissue penetration of nanocapsules.

**In vivo cancer therapy and biosafety evaluation.** Synergistic chemotherapy (i.e. phototherapy and chemo-dynamic therapy) of DOX–ICG@Fe/FeO–PPP–FA nanocapsules was investigated by treatment of solid tumors in vivo. Figure 6a shows the schematic illustration of the cancer therapy process. When laser irradiation is applied to DOX–ICG@Fe/FeO–PPP–FA nanocapsules injected

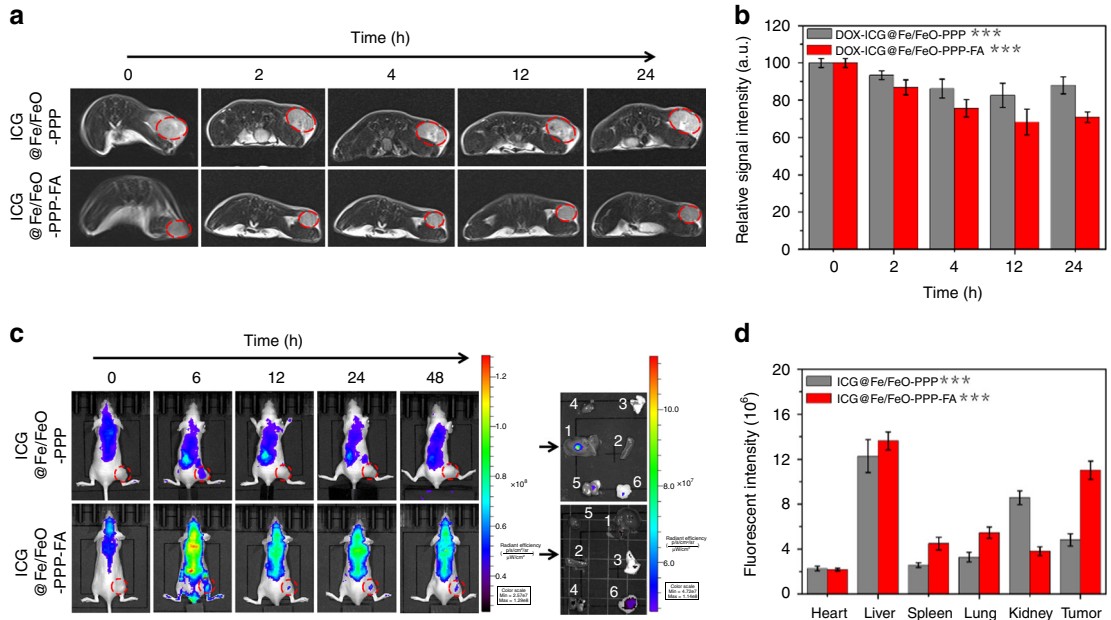

**Fig. 5** In vivo imaging. **a** Real-time MRI of KB tumor-bearing mice after intravenous injection of ICG@Fe/FeO–PPP nanocapsules and ICG@Fe/FeO–PPP–FA nanocapsules. **b** The relative MRI signal intensities changing at the tumor site after intravenous injection of ICG@Fe/FeO–PPP nanocapsules and ICG@Fe/FeO–PPP–FA nanocapsules, respectively. **c** Real-time fluorescence images of tumor-bearing mice after intravenous injection of ICG@Fe/FeO–PPP nanocapsules and ICG@Fe/FeO–PPP–FA nanocapsules. Ex vivo fluorescence images of liver (1), spleen (2), lung (3), heart (4), kidney (5) and tumor (6), which were obtained at 48 h post-injection. **d** The fluorescence intensities of the major organs after intravenous injection of ICG@Fe/FeO–PPP nanocapsules and ICG@Fe/FeO–PPP–FA nanocapsules, respectively. P values in **b** and **d** were calculated by Tukey's post-hoc test (**P < 0.01, ***P < 0.001) by comparing ICG@Fe/FeO-PPP nanocapsules with ICG@Fe/FeO–PPP–FA nanocapsules. Error bars, mean ± SD ($n = 5$)

mice, the local temperature of the tumor site rapidly increases from 37 °C to 50.4 °C within 5 min, but for the mice treated with DOX–ICG@Fe/FeO–PPP nanocapsules, the temperature only reaches to 45.2 °C (Fig. 6b and Supplementary Fig. 28). These results again confirmed the superior targeting capability of DOX–ICG@Fe/FeO–PPP–FA nanocapsules as proved in the bioimaging process. Furthermore, the bio-distribution of nanocapsules after intravenous injection for 3 days was detected by ICP-MS and confirmed the targeting capacity of DOX–ICG@Fe/FeO–PPP–FA nanocapsules in vivo (Supplementary Fig. 29). Comparing with the other six groups (ICG@Fe/FeO–PPP–FA nanocapsules and laser irradiation, ICG@Fe/FeO–PPP nanocapsules and laser irradiation, only DOX-ICG@Fe/FeO–PPP–FA nanocapsules, only ICG@Fe/FeO–PPP–FA nanocapsules, saline and laser irradiation, and only saline), the excellent antitumor efficiency of DOX–ICG@Fe/FeO–PPP–FA nanocapsules was demonstrated by tumor volume with significant inhibition and elimination in vivo (Fig. 6c). The growth status of representative nude mice in each group at the time interval of 0, 3, 6, 9, 12, 15, 18 days throughout the treatment cycle was observed (Fig. 6e). The tumor of harvested mice injected with only DOX–ICG@Fe/FeO–PPP–FA nanocapsules under the laser irradiation (808 nm, 0.3 W cm$^{-2}$) was completely eradicated after treatment. An obvious damage was evidenced to the tumor cells of mice by cell necrosis and apoptosis in the group of injection with DOX–ICG@Fe/FeO–PPP–FA nanocapsules after laser irradiation. Mice treated with other groups showed less necrotic areas (Fig. 6f and Supplementary Fig. 30). These results showed that DOX–ICG@Fe/FeO–PPP–FA nanocapsules were efficient as targeting nanomaterials with antitumor capacity in KB-bearing mice models.

Subsequently, toxicity analysis of these nanocapsules was investigated. There was no decrease in the weight of the mice in each group during the treatment which demonstrates the low toxicity of the ICG@Fe/FeO–PPP–FA nanocapsules (Fig. 6d). As

displayed in Supplementary Fig. S31, no significant difference in the levels of these liver and kidney function indicators, including alanine aminotransferase (ALT), aspartate aminotransferase (AST), alkaline phosphatase (ALP), creatinine (CRE) and blood urea nitrogen (BUN), between the treatment and control groups was observed. These results indicate the good hepatic and kidney safety profile of each group. Finally, the histological analysis was done by hematoxylin and eosin (H&E) staining of the main organs after the treatment to study the damage in acute and chronic stages. No tissue necrosis was observed in the main organs (heart, liver, spleen, lung and kidney) for the seven groups, demonstrating that the formulations mentioned above have no obvious biological toxicity (Supplementary Fig. S32).

## Discussion

In summary, we have constructed an intelligent NIR/TME dual-responsive nanocapsule (made of DOX-ICG@Fe/FeO-PPP) for enhanced tumor accumulation and improved therapy efficacy. The large initial size of these nanocapsules ensures the circulatory stability in the blood while, under irradiation of an NIR laser, the shrinkage and decomposition of nanocapsule in acidic TME guarantees intratumoral permeability of NPs and the controllable release of DOX. Interestingly, the overproduced ROS by synergistic catalysis of Fenton reaction based on Fe/FeO NCs and light activation from ICG relieves the hypoxia for solid tumors, which is necessarily required to mitigate the hypoxia-related resistance during chemo/photo- and chemodynamic therapy. As a result of these unique properties of nanocapsules, almost complete destruction of tumors was realized. In addition, dual-mode MRI and fluorescence imaging provide complementary imaging information. Hence, this study presents the design of smart nanocapsules with enhanced tumor accumulation, highly effective therapy and diagnosis to accelerate exploitation and clinical translation of intelligent theranostics nanocapsules.

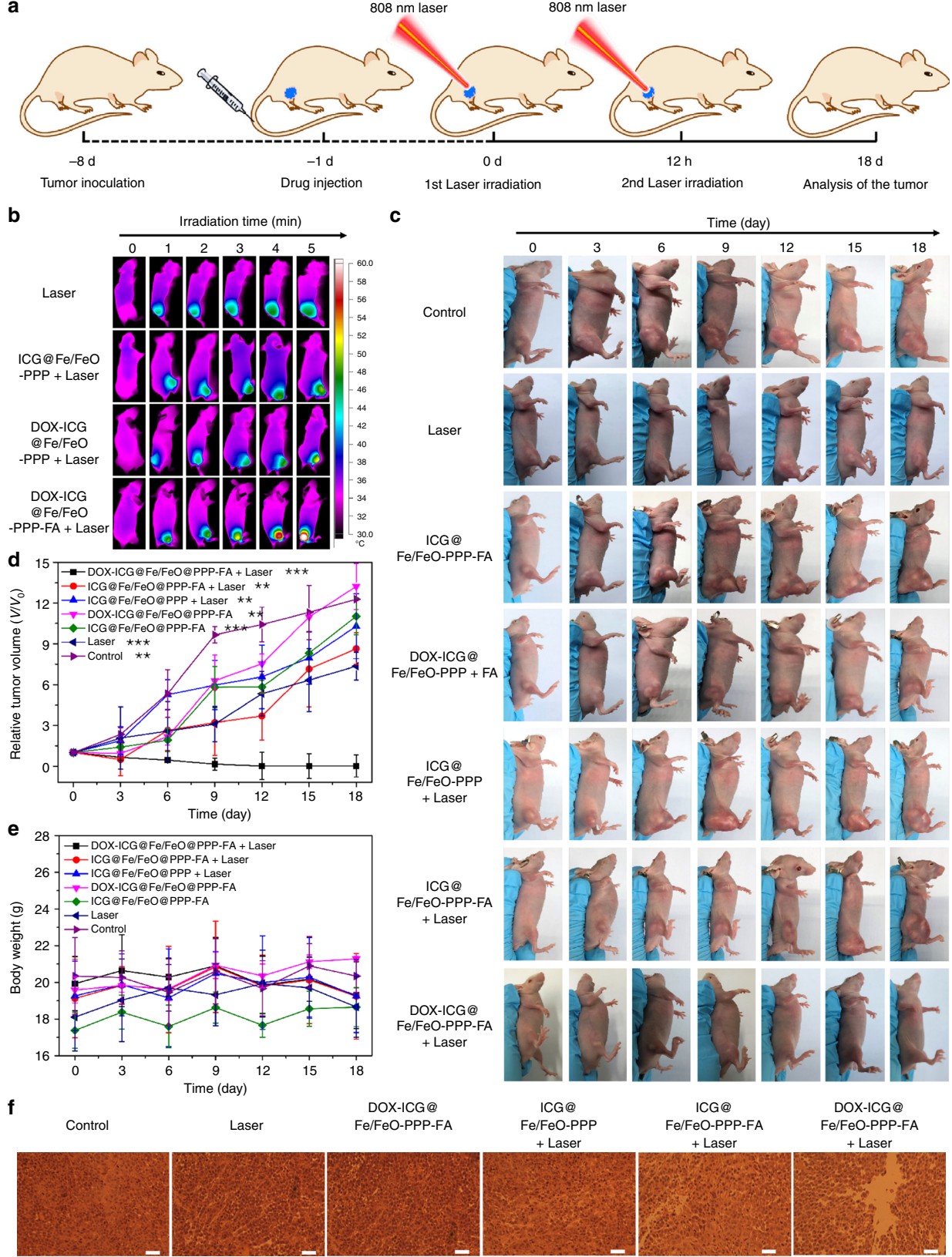

**Methods**

**Materials.** Copper (II) chloride dehydrate (CuCl₂•2H₂O, 99.9%), potassium bromide (NH₄Br, 99%) and TA (99.9%) were purchased from J&K Chemicals. Oleylamine (OAm, tech. 70 %), oleic acid (OA, 99%), poly(*N*-isopropylacrylamide) (PNIPAM, carboxylic acid terminated, Mr = 5000), dicyclohexylcarbodiimide (DCC), *N*-hydroxysuccinimide (NHS), FA, DPBF, DMP, DMPO, calcein-AM and

propidium iodide (PI) were purchased from Sigma-Aldrich. Iron pentacarbonyl was from Tianyi Co. Ltd, Jiangsu, China. Dihydrorhodamine 123 (DHR123) was purchased from Thermo Fisher Scientific (Waltham, MA, USA). H₂O₂ was obtained from Beijing Chemicals Inc. (Beijing, China). Octadecene (ODE, tech. 90%) was purchased from Alfa Asear. All the chemicals were used without additional purification, except dimethyl sulfoxide (DMSO), CHCl₃, triethylamine, and

**Fig. 6** Therapeutic effect of DOX–ICG@Fe/FeO–PPP–FA nanocapsules. **a** Schematic illustration of DOX–ICG@Fe/FeO–PPP–FA nanocapsules-based tumor therapy. **b** Real-time thermal IR images of KB tumor-bearing mice after intravenous injection of DOX–ICG@Fe/FeO–PPP nanocapsules and DOX–ICG@Fe/FeO–PPP–FA nanocapsules under 808 nm laser irradiation (0.3 W cm$^{-2}$, 5 min). **c** Volume change of tumor in the different treatment (five mice per group). **d** Body weight change of mice in the different treatment (five mice per group). **e** Representative photograph of excised tumors from euthanized mice at 18 days post treatment (five mice per group). **f** H&E-stained images of tumor regions with different treatments after 2 days of intravenous injection (scale bars: 50 μm). $P$ values in **c** were calculated by Tukey's post-hoc test (**$P < 0.01$, ***$P < 0.001$) by comparing other groups with the last group (DOX–ICG@Fe/FeO–PPP–FA + Laser). Error bars, mean ± SD ($n = 5$)

were performed under argon utilizing a homemade heating apparatus, four-necked bottles (Synthware), a glove box (MIKROUNA) and Ar/N$_2$/vacuum lines. H$_2$N–PEG–NH$_2$ (Mr = 4000), PEG-3,4-dihydroxy benzyl amine (DIB-PEG-NH$_2$) were synthesized according to oxyamination and aldimine condensation[60–62]. All the dialysis bags (Mr = 8000–14,000) were obtained from Shanghai Med.

**Instruments**. Varian 400 MHz NMR was used to acquire $^1$H-NMR spectra. The conventional bright-field images were obtained utilizing FEI Tecnai T20 microscope (200 kV), and HRTEM was carried out on an FEI Tecnai F30 microscope (300 kV). Reinforced carbon membrane support grid was used to obtain the EDS mapping. XPS measurements were performed on an imaging X-ray photoelectron spectrometer using Al Kα radiation (Axis Ultra DLD, Kratos Analytical Ltd.). All the collected spectra were calibrated with contaminated C 1s peak at 284.8 eV, and were analyzed using CasaXPS software (2.3.12 Dev7). PANalytical X'Pert3 powder X-ray diffractometer was used to obtain X-ray diffraction (PXRD) patterns with Cu-Kα ($\lambda = 0.15405$ nm) radiation at 40 kV and 40 mA. Nicolet FTIR spectrometer (Magna-IR 750) was used for FTIR measurements. Physical property measurement system (PPMS-9, Quantum Design, USA) was used to measure the magnetization. EPR spectra were measured by Bruker A300-9.5/12 spectrometer at room temperature. Fluorescent microscope (Leica DR) equipped with a digital camera (ORCA-ER, Hamamatsu) was used to obtain fluorescence microscopic images. Inductively coupled plasma-atomic emission spectrometer (ICP-AES, Prodigy 7, and Leeman, USA) was used to quantify the concentrations of Fe. Dynamic light scattering (DLS) was measured using a particle size analyzer (Zetasizer Nano ZS-90, Malvern, England). UV 1750 spectrophotometer (Shimadzu, Japan) was used to measure the UV-vis absorbance. Infrared thermal imaging instrument (FLIR A325SC camera) was used to record the temperature detection and thermal image; 808 nm high-power multimode pump laser (Shanghai Connect iber Optics Co.) was used for NIR laser.

**Synthesis of PLGA-FA**. PLGA–FA was synthesized by the amidation between PLGA and ethylenediamine-derivatized FA (FA-NH$_2$). Typically, PLGA (15000.0 mg, 0.1 mmol) and FA-NH$_2$ (48.5 mg, 0.1 mmol) were dissolved in 50 mL of dry DMSO. Subsequently, DCC (0.492 g, 2.4 mmol) and NHS (23.0 mg, 0.2 mmol) were added. The reaction mixture was stirred for 14 h at room temperature in the dark. Finally, the reaction product was isolated and purified by recrystallization. The yield of FA-NH$_2$: 60%. MS: $m/z = 482$ [M + H]$^+$. 1H NMR (DMSO-d6, 300 MHz, δ in ppm): 186–1.99 (m, 2H, C21-H), 2.01–2.06 (m, 2H, C22-H), 2.21–2.38 (m, 2H, C25-H), 2.45 (m, 2H, C26-H), 4.34 (dd, $J = 5.4$, 9.36 Hz, 1H, C19-H), 4.49 (s, 2H, C9-H2), 6.66 (d, $J = 8.7$ Hz, 2H), 7.66 (d, $J = 8.7$ Hz, 2H), 8.65 (s, 1H).

**Synthesis of PLGA–PEG–NH$_2$**. PLGA–PEG–NH$_2$ was synthesized by the amidation between PLGA and H$_2$N–PEG–NH$_2$. H$_2$N–PEG–NH$_2$ was synthesized by oxyamination method[61,62]. Subsequently, PLGA (15000.0 mg, 0.1 mmol) and H$_2$N–PEG–NH$_2$ (400.0 mg, 0.1 mmol) were dissolved in 50 mL of dry DMSO. Subsequently, DCC (24.6 mg, 0.12 mmol) and NHS (23.0 mg, 0.2 mmol) were added. The reaction mixture was stirred for 14 h at room temperature in the dark. Finally, the reaction product was isolated and purified by recrystallization.

**Synthesis of PPP**. PPP was synthesized by the amidation between PLGA–PEG–NH$_2$ and PNIPAM. Briefly, PLGA–PEG–NH$_2$ (1900.0 mg, 0.1 mmol) and PNIPAM (500.0 mg, 0.1 mmol) were dissolved in 50 mL of dry DMSO. Subsequently, DCC (24.6 mg, 0.12 mmol) and NHS (23.0 mg, 0.2 mmol) were added to the system. The reaction mixture was stirred for 14 h at room temperature in the dark. Finally, the reaction product was isolated and purified by recrystallization.

**Synthesis of Fe/FeO NCs**. Fe/FeO NCs were synthesized by a facile seed-mediated growth method. Firstly, 12 nm Fe NCs were synthesized following our previously reported method[63]. In the typical synthesis, ODE (62.5 mmol), NH$_4$Br (0.1 mmol) and OAm (1 mmol) were mixed under a gentle N$_2$ flow for 1 h in a four-necked flask. Then the solution was heated to 100 °C and kept for 1 h to remove the organic impurities. Fe(CO)$_5$ (5 mmol) was injected into the reaction system when the temperature reached 180 °C and kept for 30 min. After the system cooled down to room temperature, 27 ml of acetone was added to the system. After centrifugation, the product was washed by ethanol and hexane. The obtained Fe NCs were dispersed in 2 mL of dichloromethane. Next, Fe(acac)$_3$ (0.706 g, 4 mmol),

OA (4 mL, 12.5 mmol), OAm (6 mL, 17.5 mmol) and the resulted bcc-Fe NPs were mixed in a four-necked flask. Then the solution was heated to 120 °C and kept for 1 h to remove the organic impurities. The solution was then heated to 220 °C and kept for 30 min. After 30 min, the system was heated to 300 °C and kept for 10 min under a N$_2$ blanket. The solution was cooled down to room temperature and the NPs were washed by ethanol and hexane. Finally, the Fe/FeO NCs were dispersed in hexane.

**Synthesis of nanocapsules**. DOX–ICG@Fe/FeO–PPP nanocapsules were formulated using W/O/W emulsion method. Typically, DOX (10.0 mg) and ICG (3 mg) were dissolved in 5% w/v PVA solution (2 mL) by using ultrasound for 10 min. Fe/FeO NCs (3 mg) and PPP (25.0 mg) were dissolved in 8 mL of dichloromethane for 10 min. Then, the two were mixed and kept for 10 min by using ultrasound. The O/W emulsion was then added to 5% w/v PVA solution (40 mL) to evaporate the organic solvent at room temperature for 4 h. DOX–ICG@Fe/FeO–PPP nanocapsules were obtained after centrifugation at 6300×$g$ for 10 min. These synthesized DOX–ICG@Fe/FeO–PPP nanocapsules were treated with lyophilization for further use. DOX–ICG@Fe/FeO–PPP–FA nanocapsules, ICG@Fe/FeO–PPP nanocapsules, ICG@Fe/FeO–PPP–FA nanocapsules and ICG@PPP nanocapsules were synthesized by using the same method as DOX–ICG@Fe/FeO–PPP nanocapsules, The only difference was that the ratio of PLGA–FA:PPP was 5:95 in the synthesis of DOX–ICG@Fe/FeO–PPP–FA nanocapsules and ICG@Fe/FeO–PPP–FA nanocapsules.

**Photothermal effect of ICG@Fe/FeO–PPP nanocapsules**. A total of 350 μL of ICG@Fe/FeO–PPP nanocapsule dispersions with different concentrations (0, 10, 20, 40, 60 and 80 mg L$^{-1}$) were irradiated with a laser (808 nm, 0.3 W cm$^{-2}$) for 5 min, and their temperature in solution was recorded by an online type thermocouple thermometer. Similarly, in order to study the influence of optical density on photothermal conversion, 350 μL of 60 mg L$^{-1}$ ICG@Fe/FeO–PPP nanocapsule dispersions were irradiated with an 808-nm laser with different power densities (0.2, 0.4, 0.6, 0.8 and 1.0 W cm$^{-2}$) for 5 min. The change of temperature in solution was recorded by an online type thermocouple thermometer. The photostability of ICG@Fe/FeO–PPP nanocapsule dispersions (60 mg L$^{-1}$) was estimated by irradiating in a quartz cuvette with a laser (808 nm, 0.3 W cm$^{-2}$) for 5 min and then cooling to room temperature without irradiation. The photostability was tested by repeating such processes four times.

**Cell culture**. NIH3T3 and KB cell lines were obtained from the Cancer Institute and Hospital of the Chinese Academy of Medical Science. All cell-culture-related reagents were purchased from Invitrogen. RPMI-1640 culture medium supplemented with 10% FBS and 1% penicillin/streptomycin was used to culture cells at 37 °C under 5% CO$_2$ with 100% humidity.

**In vitro cytotoxicity assay**. NIH3T3 or KB cells (1 × 104 cells per well) seeded into a 96-well cell culture plate were incubated with ICG@Fe/FeO–PPP nanocapsules in different Fe concentrations (0, 0.01, 0.10, 0.20, 0.40, 0.80, 1.60 and 3.20 mM) for 48 h at 37 °C under 5% CO$_2$. The relative cell viabilities were determined by a standard CCK-8 viability assay ($n = 3$).

**In vitro photothermal ablation of KB cells**. NIH3T3 and KB cells (1 × 10$^4$ cells per well) seeded into a 96-well cell culture plate were incubated with ICG@Fe/FeO–PPP–FA nanocapsules, DOX–ICG@Fe/FeO–PPP–FA nanocapsules, ICG@Fe/FeO–PPP nanocapsules and laser irradiation, ICG@Fe/FeO–PPP–FA nanocapsules and laser irradiation, DOX–ICG@Fe/FeO–PPP–FA nanocapsules and laser irradiation in Fe concentrations of 0, 0.01, 0.10, 0.20, 0.40, 0.80, 1.60 and 3.20 mM for 24 h at 37 °C under 5% CO$_2$, respectively. The cells were washed three times with PBS and fed with fresh medium, followed by irradiating with laser for 5 min (808 nm, 0.3 W cm$^{-2}$), respectively. Finally, the viability of cells was evaluated by a standard CCK-8 assay ($n = 3$).

To examine the photothermal effect of ICG@Fe/FeO–PPP–FA nanocapsules on KB cells in vitro, KB cells seeded (1 × 10$^4$ cells per well) in culture dishes were incubated with ICG@Fe/FeO–PPP–FA nanocapsules and laser irradiation, ICG@Fe/FeO–PPP nanocapsules and laser irradiation, DOX–ICG@Fe/FeO–PPP–FA nanocapsules only, ICG@Fe/FeO–PPP–FA nanocapsules only, laser irradiation only and control (without any treatment) for 4 h, respectively. Laser

(808 nm, 0.3 W cm$^{-2}$) was used to irradiate the adherent cell solution. After DMEM was removed, the cells were washed with PBS three times. Calcein-AM (100 μL) and PI solution (100 μL) were incubated with KB cells for 15 min. Living cells were stained with calcein-AM (green fluorescence) and dead cells with PI (red fluorescence) solution ($n = 3$).

**In vitro •OH generation of ICG@Fe/FeO–PPP nanocapsules**. The •OH generated by ICG@Fe/FeO–PPP nanocapsules was detected by TA oxidation method. The whole process was based on the fluorescence spectrum because after the oxidation of TA to TAOH by •OH, nonfluorescent TA was converted to fluorescent TAOH. The experiment was divided into four groups: (1) only under 808 nm laser irradiation (0.3 W cm$^{-2}$), (2) ICG@PPP nanocapsules under 808 nm laser irradiation (0.3 W cm$^{-2}$), (3) Fe/FeO under 808 nm laser irradiation (0.3 W cm$^{-2}$) and (4) ICG@Fe/FeO–PPP nanocapsules under 808 nm laser irradiation (0.3 W cm$^{-2}$). Firstly, TA solution was prepared (1 mM); subsequently, different groups were treated accordingly, and the fluorescence spectra at different times (0, 5, 10, 15, 20, 25 and 30 min) were measured (ex/em = 327 nm/437 nm). Laser irradiation time was 5 min ($n = 6$).

**In vitro $^1O_2$ generation of nanocapsules**. The $^1O_2$ generation potential of all nanomaterials in vitro was detected by using DPBF. Briefly, 0.5 mL of the sample suspension ($C_{Fe} = 0.10$ mM, $C_{nanocapsules} = 60$ mg L$^{-1}$) was added to 0.5 mL of DPBF (0.02 mM) solution (phosphate buffer, pH 5.4). The mixture was irradiated by laser (808 nm, 0.3 W cm$^{-2}$) for 5 min ($n = 6$). The concentration of DPBF was determined by measuring the absorbance at 466 nm as a function of time using a UV-vis absorption spectra. The rate constant for $^1O_2$ generation by ICG@PPP nanocapsules, Fe/FeO NCs and ICG@Fe/FeO–PPP nanocapsules was calculated using the following Eq. (4):

$$\ln\left(\frac{C_t}{C_0}\right) = \ln\left(\frac{A_t}{A_0}\right) = -kt \qquad (4)$$

where $C_t$ is the concentration of DPBF at a certain time point; $C_0$ is the concentration of DPBF at the initial time point; $A_t$ is the absorbance at the wavelength of 466 nm at a certain time point; $A_0$ is the absorbance at the wavelength of 466 nm at the initial time point; $k$ is the reaction rate constant, which was calculated by correlated fitting linear equation;and $t$ is the mixing time for DPBF and each nanomaterial.

**Evaluation of ROS generation in KB cells**. ROS-generating capabilities of free ICG and ICG@Fe/FeO–PPP–FA nanocapsules under normoxic or hypoxic condition were assessed by DHR123, respectively[11]. DHR123 staining was carried out as follows: cells were incubated with ICG (0.05 mM) or ICG@Fe/FeO–PPP–FA nanocapsules (60 mg L$^{-1}$) for 24 h. Then, 1 μg of DHR123 was added to cell media under laser irradiation (808 nm, 0.3 W cm$^{-2}$) for 1 min. Confocal laser scanning microscopy (CLSM) and flow cytometry were used to observe the fluorescence intensity by DHR123 (ex/em = 488 nm/520 nm, $n = 3$).

**DOX delivery evaluation of DOX–ICG@Fe/FeO–PPP nanocapsules**. 5 mg of DOX–ICG@Fe/FeO–PPP nanocapsules DOX loading and releasing of DOX–ICG@Fe/FeO–PPP nanocapsules. A total of 5 mg of DOX–ICG@Fe/FeO–PPP nanocapsules were added in 10 mL of dichloromethane under ultrasound for 1 h. The absorbance at the wavelength of 481 nm for the DOX from the decomposed DOX–ICG@Fe/FeO–PPP nanocapsules was measured by UV-vis absorption spectra ($n = 6$). Loading of DOX in DOX–ICG@Fe/FeO–PPP nanocapsules was quantified using the simulation of standard working curve for the detection of DOX. Loading capacity was calculated using the following Eq. (5):

$$\text{Encapsulation efficiency (\%)} = \frac{M_{\text{DOX loaded}}}{M_{\text{nanocapsules}}} \times 100\% \qquad 5$$

where $M_{\text{DOX loaded}}$ is the mass of DOX detected after the treatment of dichloromethane and $M_{\text{nanocapsules}}$ is the total mass of nanocapsules.

The release of DOX from DOX–ICG@Fe/FeO–PPP nanocapsules was evaluated by dialyzing the nanocapsules in the dark in phosphate buffer at pH 7.4, 6.5 and 5.4 for different times at 37 °C in 80 h. UV-vis absorption spectra were used to determine the released DOX at the wavelength of 481 nm. For the 808 nm laser-triggered DOX release, DOX–ICG@Fe/FeO–PPP nanocapsules were dialyzed in the buffer solution with pH values of 7.4, 6.5 or 5.4 at 37 °C. Released DOX was collected at the time points of 1, 2, 3, 4 and 5 h. At the time periods of 1, 2, 3, and 4 h, a laser (808 nm, 0.3 W cm$^{-2}$) was employed for 5 min. The released DOX was collected immediately and the dosage was measured using simulation of standard working curve for the detection of DOX. In addition, the release of DOX from DOX–ICG@PPP nanocapsules was followed as the same method above ($n = 6$).

**Animals and tumor model**. All experiments involving animals were performed in accordance with the guidelines of the Institutional Animal Care and Use Committee (IACUC) of Peking University, Beijing, China. Four- to five-week-old Balb/c nude mice with the average weight of 20 g were provided by the Beijing Center for Disease Control and Prevention, Beijing, China. SPF animal house was provided to mice under a 12-h light and 12-h darkness cycle and were fed a standard laboratory diet and tap water ad libitum. KB cells (0.2 mL cells in 1640 culture medium without FBS) were injected into the mice subcutaneously at the right axillary region.

**In vivo MRI**. For MRI in vivo, the KB-tumor-bearing mice were intravenously injected with ICG@Fe/FeO–PPP nanocapsules and ICG@Fe/FeO–PPP–FA nanocapsules (20 mg kg$^{-1}$, 200 μL). After the injection, $T_2$ images were obtained at 0, 6, 12, 24 and 48 h by a clinic 3T MRI scanner (Philips, TR = 1200 ms, TE = 30.2 ms, slice thickness = 2.5 mm). The intensity of the MRI signal before injection was used as the control ($n = 5$).

**In vivo fluorescence imaging**. The KB-tumor-bearing mice were intravenously injected with ICG@Fe/FeO–PPP nanocapsules and ICG@Fe/FeO–PPP–FA nanocapsules (20 mg kg$^{-1}$, 200 μL) for fluorescence imaging in vivo. The fluorescence signal was recorded by the CRi maestro ex in vivo imaging system (USA) at 0, 6, 12, 24 and 48 h after the injection. The fluorescence signal before injection was used as the control. To confirm the in vivo distribution of ICG@Fe/FeO–PPP nanocapsules and ICG@Fe/FeO–PPP–FA nanocapsules, mice were sacrificed 48 h post-injection. The main organs (liver, heart, lung, spleen, tumor and kidneys) were collected for imaging and semi-quantitative biodistribution analysis ($n = 5$).

**In vivo antitumor efficiency evaluation**. Mice bearing 200 mm$^3$ KB tumors were randomly divided into seven groups: (1) DOX–ICG@Fe/FeO–PPP–FA nanocapsules and laser irradiation; (2) ICG@Fe/FeO–PPP–FA nanocapsules and laser irradiation; (3) ICG@Fe/FeO–PPP nanocapsules and laser irradiation; (4) only DOX–ICG@Fe/FeO–PPP–FA nanocapsules; (5) only ICG@Fe/FeO–PPP–FA nanocapsules; (6) saline and laser irradiation and (7) control (only saline). Five mice were contained in each group. After 200 mL of saline or nanocapsules (20 mg kg$^{-1}$) were intravenously injected into nude mice bearing the KB tumor for 24 h, mice were exposed to 808 nm laser (0.3 W cm$^{-2}$) for 5 min (Fig. 5a) at the first time. Subsequently, the second irradiation by 808 nm laser (0.3 W cm$^{-2}$, 5 min) was started after 12 h. The changes in body weight and tumor volume during 18 days of treatment period were recorded ($n = 5$).

**In vivo blood biochemistry test**. Mice were randomly divided into five groups: (1) DOX–ICG@Fe/FeO–PPP–FA nanocapsules and laser irradiation; (2) ICG@Fe/FeO–PPP–FA nanocapsules and laser irradiation; (3) ICG@Fe/FeO–PPP nanocapsules and laser irradiation; (4) only DOX–ICG–Fe/FeO–PPP–FA nanocapsules and (5) control (only saline). After 200 mL of nanocapsules or saline (20 mg kg$^{-1}$) were intravenously injected into nude mice for 72 h, the blood biochemistry test was started, which included five important hepatic and kidney function indicators (ALT, AST, ALP, CRE and BUN) ($n = 5$).

**Histological evaluation**. Mice from each group were euthanized, then major organs and tumor were recovered, followed by fixing with 10% neutral buffered formalin after 18 days treatment. The organs were embedded in paraffin and sectioned at 5 mm; H&E or Prussian blue staining was performed for histological examination. The slides were observed under an optical microscope ($n = 5$).

**Statistical analysis**. Statistical analysis was calculated by Tukey's post-hoc test with statistical significance assigned at **$P < 0.01$ (moderately significant), ***$P < 0.001$ (highly significant).

**Ethical approval**. All experiments involving animals were performed in accordance with the guidelines of the Institutional Animal Care and Use Committee (IACUC) of Peking University, Beijing, China.

Original data are provided in a Source Data file.

**Reporting summary**. Further information on experimental design is available in the Nature Research Reporting Summary linked to this paper.

## Data availability
The source data underlying Figs. 2e, f, 3a, b, 4a–c, 5b, d, 6c, d and Supplementary Figs. 2a, b, 5a, b, 6a–d, 7a–f, 8a, b, 9a–c, 10a–d, 12a, b, 13a, b, 17a, b, 18, 19a–d, 21a–d, 24a, b, 25, 26, 27b, 28, 29 and 31 are provided as a Source Data file. All the other data supporting the findings of this study are available within the article and its Supplementary Information files and from the corresponding authors upon reasonable request. A reporting summary for this article is available as a Supplementary Information file.

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

## Acknowledgements

This work was financially supported by the Beijing Municipalit Unicipality of Natural Science Foundation (L72008), the National Natural Science Foundation of China (51672010, 21671088, 81421004, 21431002, 51631001, 51590882, 21575161), the National Key R&D Program of China (2017YFA0206301, 2016YFA0200102), and the Key Laboratory of Biomedical Effects of Nanomaterials and Nanosafety, National Center for Nanoscience and Technology, Chinese Academy of Sciences (NSKF201607). Thanks for the help of the radial stress analysis for nanocapsules in the shrinking process from Fuyao Zhao.

## Author contributions

B.W. and Y.H. conceived and designed the experiments. Z.W., Y.J., H.Y., F.S. and J.L. performed the experiments. Z.W., B.W. and Y.H. analyzed the results. Z.W., Y.J., Z.A., B.W. and Y.H. wrote the manuscript.

## Competing interests

The authors declare no competing interests.
