## [Peer Review File · Nature Communications]

Reviewers' comments:

Reviewer #1 (Remarks to the Author):

This manuscript reported the synthesis and preparation of near-infrared (NIR) light-/tumor microenvironment (TME)- dual responsive size-switchable nanocapsules, which not only shows NIR light and TME responsive size changes for enhanced tumor accumulation, but also can regulate the unfavorable TME for improving the synergistic therapy effect of cancer. The synthesis and nanocapsules structure is relatively new. The characterizations are overall comprehensive and the results have good scientific value and novelty. Therefore, I suggest the acceptance for publication after the following points are addressed:

1. In this manuscript, for the first time, the authors synthesized nanocapsules that could simultaneously perform photodynamic and chemodynamic therapy. Because both ROS treatment modes need to consume hydrogen peroxide, is there competition in the treatment process?
2. In the ESR spectra of Fe/FeO NCs and ICG@Fe/FeO-PPP nanocapsules with DMPO, the author should give the physical parameters represented by abscissa.
3. Can Fe/FeO NCs catalyze the release of oxygen from hydrogen peroxide in tumor microenvironment?
4. In order to explain the formation and assembly process of nanocapsules more clearly, the authors should describe the infrared spectra in detail in the supporting information.
5. What is the advantage of Fe/FeO NCs in the nanocapsules?

Reviewer #2 (Remarks to the Author):

Dear Editor of Nature Communications,

After reading carefully the manuscript entitled "Near-Infrared Light and Tumor Microenvironment Dual Responsive 2 Size-Switchable Nanocapsules for Multimodal Tumor Theranostics" by Zhiyi Wang, Yanmin, Zeeshan Ali, Hui Yin, Fugeng Sheng, Jian Lin, Baodui Wang and Yanglong Hou, I would not suggest its publication in Nature Communications.

The reason of my decision is due to the lack of novelty of the presented work. In this manuscript, the authors prepare a multifunctional system that can be used for diagnosis and cancer treatment, and in order to do that the combine many different elements like magnetic nanoparticles, block copolymer (BTW, the FDA has approved PLGA for some uses but PNIPAM, the thermo-responsive polymer is not approved), DOX and ICG as building blocks for creating hybrid nano-capsules of 200 nm in diameter. All the experiments are well described and many of the results sounds, other like the EPR capacity of 200 nm nanocapsules is a fairy tale.

Furthermore, a theoretical pH value of 5.4 within solid tumour is not possible, (this value is not compatible with the cell viability). However, the main problem of this manuscript is that there are many other multifunctional systems that have been already published and many of them combine similar elements to produce a theragnostic system like the presented in this work. In addition, the presented manuscript isn't a break-through in the topic, is a good manuscript that present nice results for being published elsewhere like ACS applied materials and interfaces or Nano-Research, but not in Nature Communication.

Best regards

Reviewer #3 (Remarks to the Author):

This study reports a novel kind of near-infrared (NIR) light and tumor microenvironment (TME) dual responsive size-switchable nanocapsules (DOX-ICG@Fe/FeO-PPP). The smartly engineered DOX-ICG@Fe/FeO-PPP nanocapsules were shown to not only shrink and decompose into small-sized nanodrugs upon drug release, but also regulate TME to overproduce reactive oxygen species (ROS) for enhanced synergistic therapy in tumors. Sufficient experiments were performed to verify the magic theranostic performance of DOX-ICG@Fe/FeO-PPP nanocapsules. This paper is also well-written and well-organized. Therefore, I would like to recommend this paper for publication in Nature Communications after addressing the following issues:

1. The mechanisms for the synthesis of the PLGA-PEG-PNIPAM (PPP) polymer and construction of DOX-ICG@Fe/FeO-PPP nanocapsules are suggested to be introduced in detail.
2. The polydispersity index (PDI) values for the DLS results in Supplementary Figure S5 should be given.
3. Where is the source of $1O_2$? Why does the $1O_2$ generation rely on pH? It seems that the Fenton reaction of ICG@Fe/FeO-PPP can only generate $\bullet OH$ rather than $1O_2$.
4. The generated $\bullet OH$ amount should be measured by using more accurate quantitative methods, such as methylene blue (MB) bleaching (Angew. Chem. Int. Ed. 2015, 54, 1770), terephthalic acid (TA) oxidation (Nano Lett. 2017, 17, 4323).
5. To validate the confocal fluorescence imaging result in Figure 3d, the intracellular ROS must also be measured through flow cytometry analysis.
6. The authors attribute the increased drug release of DOX-ICG@Fe/FeO-PPP nanocapsules to the instability of Fe/FeO NCs in weak acidic condition. Why was Fe/FeO NCs unstable in weak acidic condition? Can the authors provide convincing data?
7. There is no data to show the advantage of laser-triggered shrinkage of DOX-ICG@Fe/FeO-PPP nanocapsules in facilitating the deep tumor tissue penetration of nanocapsules.
8. Some in vitro cell experiment results are suggested to be removed from supplementary information to article. Also, please add statistical analysis to Figure 4a, 4b, 5d, 6c, S16b, S20.
9. The quality of the figures should be improved. Currently the words in the figures are not shown clearly.

Response to the comments from the editor and three referees

Response to the editor

The comments:

The reports seem to be quite clear, and we will naturally need you to carefully address all of the issues raised. We would recommend that you expand your discussion of similar size changing systems in the literature and highlight the differences to your system. In addition we would ask that you carefully consider the laser power being used in your experiments. The current safety standard is 0.3 W/cm², as a minimum we need you to confirm that the ethical approval for your animal work approved the use of the higher laser powers.

Our response and revision:

Thanks for your precious comments, evaluation and publication recommendation of our manuscript. The “size changing systems” reported in our study is quite different from already reported systems. It is well known that the design and development of **sophisticated nanocapsules** for targeted delivery of theranostic agents to solid tumors hold **great promise** for improving the treatment efficacy and minimizing the systemic toxicity (*J. Am. Chem. Soc.* 2019, 141, 4406-4411; *J. Am. Chem. Soc.* 2017, 139, 4584-4610; *Macromolecules*, 2013, 46, 9169-9180). Despite tremendous potential, currently most researches who studies nanocapsules like structures, have mainly focused on a **single** stimuli responsive system, which does not meet precise control and initiation of such responsive systems (*Angew. Chem. Int. Ed.* 2019, 58, 159-163; *Angew. Chem. Int. Ed.* 2018, 57, 17048-17052; *J. Am. Chem. Soc.* 2017, 139, 7522-7532; *J. Am. Chem. Soc.* 2014, 136, 14896-14902; *Mol. Pharmaceutics* 2014, 11, 1599-1610). Moreover, the **metabolism** of nanocapsules in biological environment has **rarely** been studied (*J. Am. Chem. Soc.* 2018, 140, 4666-4677; *Langmuir* 2016, 32, 6211-6225). Unfortunately, most of the existing nanocapsules **lack** the simultaneous combination of the **multiple therapeutic/imaging modalities** into one platform to synergistically enhance

therapeutic efficacy or to obtain theranostic nanomaterials (*ACS Appl. Mater. Interfaces* 2019, 11, 1886-1895; *Acc. Chem. Res.* 2015, 48, 2935-2946). **More importantly**, the toxicity of most of the materials used to construct nanocapsules is not approved by FDA. Therefore, it is **difficult** for these nanocapsules to enter **clinical trials**.

In our study, we have constructed an intelligent NIR/TME dual-responsive nanocapsule (made of DOX-ICG@Fe/FeO-PPP) for enhanced tumor accumulation and improved therapy efficacy. This research work includes the following novelty:

1. Under the **stimulation** of NIR light and acid TME, the nanocapsules could shrink and decompose into **small-sized** nano-drugs, accompanied by drug release.
2. Meanwhile, the nano-drugs enter into tumors and overproduce the reactive oxygen species (ROS) by synergistic catalysis of Fenton reaction based on Fe/FeO NCs and light activation from ICG, which relieves hypoxic condition and promotes the **synergistic** therapy of tumors.
3. *In vivo* experiments demonstrated that these nanocapsules can offer **remarkable** imaging and therapeutic results.
4. The materials that make up nanocapsules are basically FDA **certified** (PLGA, PEG and ICG). The furthermore experimental results show that the **biosafety** of the produced nanocapsules is relatively **reliable**.

Hence, this study presents the design of smart nanocapsules with enhanced tumor accumulation, highly effective therapy and diagnosis to **accelerate** exploitation and **clinical translation** of intelligent theranostics` nanocapsules.

To ensure that our research work conforms to the ethical approval for our animal work, we reduced the laser power to 0.3 W/cm². In addition, we improved the encapsulation of ICG by PLGA-PEG-PNIPAM (The coating amount changed from 2 mg to 3 mg). Finally, the feasibility of this laser power (808 nm, 0.3 W/cm²) has been

proved both in vitro and in vivo. Following experimental results were included in revised manuscript:

Figure 4. | DOX delivery evaluation of DOX-ICG@Fe/FeO-PPP nanocapsules. (a) DOX release curves of DOX-ICG@PPP nanocapsules and DOX-ICG@Fe/FeO-PPP nanocapsules in different pH value (7.4, 6.5 and 5.4) at 37 °C (**P<0.01, ***P<0.001). (b) DOX release curves of DOX-ICG@PPP nanocapsules (pH=7.4, 6.5 and 5.4) and DOX-ICG@Fe/FeO-PPP nanocapsules (pH=7.4, 6.5 and 5.4) with four laser on/off cycles under an laser irradiation (808 nm, 0.3 W cm⁻²) for 5 min (**P<0.01, ***P<0.001). (c) TEM image of the shrinking process for DOX-ICG@Fe/FeO-PPP nanocapsules in 48 h after the irradiation of laser (808 nm, 0.3 W cm⁻²) for 5 min (pH 6.5). (d) Schematic diagram of radial stress analysis of DOX-ICG@Fe/FeO-PPP nanocapsules in the shrinking process, and P1, P2 and P3 were represent the radial stress for the nanocapsules in different stages respectively. (e) Schematic diagram of the degradability for PPP in the pH below 6.5.

Supplementary Figure S19. (a) UV-vis absorbance spectra of ICG@Fe/FeO-PPP nanocapsules. (b) Temperature curves of ICG@Fe/FeO-PPP nanocapsules dispersions with concentrations of 0, 20, 40, 60 and 80 mg L⁻¹ under 808 nm irradiation at a laser power density of 0.3 W cm⁻² in 5 min. (c) Temperature curves of ICG@Fe/FeO-PPP nanocapsules dispersions with different power densities (0.1, 0.2, 0.3, 0.4, and 0.5 W cm⁻²) with the concentration of 60 mg L⁻¹ under 808 nm laser irradiation in 5 min.; (d) Temperature curves of ICG@Fe/FeO-PPP nanocapsules for four laser on/off cycles under the 808 nm laser with 0.3 W cm⁻². TEM image of ICG@Fe/FeO-PPP nanocapsules before (e) and after (f) 5 min irradiation of 808 nm laser in the concentration of 60 mg L⁻¹.

Figure 6. | Evaluation of cancer therapeutic effect for DOX-ICG@Fe/FeO-PPP-FA nanocapsules.

(a) Schematic illustration of DOX-ICG@Fe/FeO-PPP-FA nanocapsules-based tumor therapy. (b) Real-time thermal IR images of KB tumor-bearing mice after intravenous injection of DOX-ICG@Fe/FeO-PPP nanocapsules and DOX-ICG@Fe/FeO-PPP-FA nanocapsules under 808 nm laser irradiation (0.3 W cm^{-2} , 5 min). (c) Volume change of tumor in the different treatment. (d) Body weight change of mice in the different treatment (** $P < 0.01$, *** $P < 0.001$). (e) Representative photograph of excised tumors from euthanized mice at 18 days post treatment. (f) H&E stained images of tumor regions with different treatments after 2 days of intravenous injection.

Supplementary Figure S27. Temperature change curves of tumor-bearing mice at different time points in the different treatment groups (Laser: 808 nm, 0.3 W cm⁻²).

Response to the referee 1:

The comments:

This manuscript reported the synthesis and preparation of near-infrared (NIR) light-/tumor microenvironment (TME)- dual responsive size-switchable nanocapsules, which not only shows NIR light and TME responsive size changes for enhanced tumor accumulation, but also can regulate the unfavorable TME for improving the synergistic therapy effect of cancer. The synthesis and nanocapsules structure is relatively new. The characterizations are overall comprehensive and the results have good scientific value and novelty. Therefore, I suggest the acceptance for publication after the following points are addressed:

Our response:

Thanks to the respected reviewer for appreciating comments and positive evaluation on our manuscript. We have addressed all the questions of reviewer and tried our best to perform additional experiments where needed.

Q1. In this manuscript, for the first time, the authors synthesized nanocapsules that could simultaneously perform photodynamic and chemodynamic therapy. Because both ROS treatment modes need to consume hydrogen peroxide, is there competition in the treatment process?

Our response:

Thanks a lot to the reviewer's professional question. Actually, the role of photodynamic therapy in our treatment methods is relatively weak. The main purpose of ICG and laser is to promote the shrinkage of nanocapsules. The laser power used is relatively low and the consumption of hydrogen peroxide is limited.

Q2. In the ESR spectra of Fe/FeO NCs and ICG@Fe/FeO-PPP nanocapsules with DMPO, the author should give the physical parameters represented by abscissa.

Our response and revision:

Many thanks to the reviewer for pointing out this short coming. We have revised Figure 3c. Followed is the revised Figure:

Figure 3. | (c) ESR spectra of Fe/FeO NCs and ICG@Fe/FeO-PPP nanocapsules with DMPO as the spin trap in different pH value condition (7.4, 6.5 and 5.4).

Q3. Can Fe/FeO NCs catalyze the release of oxygen from hydrogen peroxide in tumor microenvironment?

Our response:

Fe/FeO NCs catalyze the release of oxygen from hydrogen peroxide in tumor microenvironment. According to our proposed reaction mechanism, Fe/FeO NCs mainly undergo the above-mentioned Fenton reaction process in the tumor microenvironment. In this condition, Fe/FeO NCs will dissociate and produce Fe³⁺ species. These Fe³⁺ have been proved to catalyze the decomposition process of

hydrogen peroxide to produce oxygen in tumor cells (*Nat. Nanotechnol.*, 2007, 2, 577-583; *Anal. Chem.*, 2012, 84, 5753-5758).

Q4. In order to explain the formation and assembly process of nanocapsules more clearly, the authors should describe the infrared spectra in detail in the supporting information.

Our response and revision:

Thanks a lot to the reviewer for this valuable suggestion. We have elaborated the infrared spectra in detail and revised the supporting information discussion on page 4 line 10-15.

“The formation of PPP was confirmed by the FTIR spectrometer, which was marked in Supplementary Figure S2. The red shift of the absorption peak for the stretching vibration of the C=O from carboxyl group (1635 cm^{-1}) to amide bond (1689 cm^{-1}) proves the amination of folic acid molecule (Supplementary Figure S2a: i and ii). The existence of vibration absorption peaks (3410 cm^{-1} and 1480 cm^{-1}) for N-H bond (Supplementary Figure S2a: iv) proved the successful synthesis of PLGA-FA. The formation of amide in PPP was also proved in Supplementary Figure S2b.”

Q5. What is the advantage of Fe/FeO NCs in the nanocapsules?

Our response:

Many thanks to the reviewer raising this question. The efficiency of Fe^{2+} in catalyzing fenton reaction is higher than that of other valence states. Hence, the core-shell structure of Fe/FeO NCs helps to obtain more Fe^{2+} in tumor microenvironment. Because the oxidation of Fe in the core is a process from low to high valence, Fe^{2+} can be directly obtained from shell ferrous oxide in weak acid tumor microenvironment. So the selection of Fe/FeO NCs is very meaningful.

Response to Referee Two

Reviewer Comments:

After reading carefully the manuscript entitled "Near-Infrared Light and Tumor Microenvironment Dual Responsive 2 Size-Switchable Nanocapsules for Multimodal Tumor Theranostics" by Zhiyi Wang, Yanmin, Zeeshan Ali, Hui Yin, Fugeng Sheng, Jian Lin, Baodui Wang and Yanglong Hou, I would not suggest its publication in Nature Communications.

The reason of my decision is due to the lack of novelty of the presented work. In this manuscript, the authors prepare a multifunctional system that can be used for diagnosis and cancer treatment, and in order to do that the combine many different elements like magnetic nanoparticles, block copolymer (BTW, the FDA has approved PLGA for some uses but PNIPAM, the thermo-responsive polymer is not approved), DOX and ICG as building blocks for creating hybrid nano-capsules of 200 nm in diameter. All the experiments are well described and many of the results sounds, other like the EPR capacity of 200 nm nanocapsules is a fairy tale.

Furthermore, a theoretical pH value of 5.4 within solid tumour is not possible, (this value is not compatible with the cell viability). However, the main problem of this manuscript is that there are many other multifunctional systems that have been already published and many of them combine similar elements to produce a theragnostic system like the presented in this work. In addition, the presented manuscript isn't a break-through in the topic, is a good manuscript that present nice results for being published elsewhere like ACS applied materials and interfaces or Nano-Research, but not in Nature Communication.

Our response and revision:

Thanks to the reviewer for critical comments and evaluation on our manuscript. It is well known that the design and development of **sophisticated nanocapsules** for targeted delivery of theranostic agents to solid tumors hold **great promise** for improving treatment efficacy and minimizing systemic toxicity (*J. Am. Chem. Soc.* 2019, 141, 4406-4411; *J. Am. Chem. Soc.* 2017, 139, 4584-4610; *Macromolecules*, 2013, 46, 9169-9180). Despite tremendous potential, currently most researches on nanocapsules are mainly focused on **single** stimuli responsive, which does not meet precise control and initiation of the responsive systems (*Angew. Chem. Int. Ed.* 2019,

58, 159-163; *Angew. Chem. Int. Ed.* 2018, 57, 17048-17052; *J. Am. Chem. Soc.* 2017, 139, 7522-7532; *J. Am. Chem. Soc.* 2014, 136, 14896-14902; *Mol. Pharmaceutics* 2014, 11, 1599-1610). Moreover, the **metabolism** of nanocapsules in biological environment has **rarely** been studied (*J. Am. Chem. Soc.* 2018, 140, 4666-4677; *Langmuir* 2016, 32, 6211-6225). Furthermore, most of the existing nanocapsules **lack** the simultaneous combination of the **multiple therapeutic/imaging modalities** into one platform to synergistically enhance therapeutic efficacy or to obtain theranostic nanomaterials (*ACS Appl. Mater. Interfaces* 2019, 11, 1886-1895; *Acc. Chem. Res.* 2015, 48, 2935-2946). **More importantly**, the toxicity of the most of the materials used to construct nanocapsules is not approved by FDA. Therefore, it is difficult for these nanocapsules to enter clinical trials.

In our study, we **for the first time** have constructed an intelligent NIR/TME dual-responsive nanocapsule (made of DOX-ICG@Fe/FeO-PPP) for enhanced tumor accumulation and improved therapy efficacy. This research work includes the following novelty:

1. Under the **stimulation** of NIR light and acid TME, the nanocapsules could shrink and decompose into **small-sized** nano-drugs, accompanied by drug release.
2. Meanwhile, the nano-drugs enter into tumors and overproduce the reactive oxygen species (ROS) by synergistic catalysis of Fenton reaction based on Fe/FeO NCs and light activation from ICG, which relieves hypoxic condition and promotes the **synergistic** therapy of tumors.
3. *In vivo* experiments demonstrated that these nanocapsules can offer **remarkable** imaging and therapeutic results.
4. The materials that make up nanocapsules are basically FDA **certified** (PLGA, PEG and ICG). The furthermore experimental results show that the **biosafety** of the produced nanocapsules is relatively **reliable**.

Hence, this study presents the design of smart nanocapsules with enhanced tumor accumulation, highly effective therapy and diagnosis to **accelerate exploitation** and **clinical translation** of intelligent theranostics nanocapsules.

In addition, we have responded to other mentioned questions and performed experiments or revisions to improve our manuscript as follows:

The reviewer pointed out that the “**block copolymer (BTW, the FDA has approved PLGA for some uses but PNIPAM, the thermo-responsive polymer is not approved)**”. This misunderstanding was caused by the sentence “It worths mentioning here that these nanocapsules has potential clinical application as the constituent materials of these have already been approved by the US FDA” in our manuscript. Actually, PEG and PLGA in the PLGA-PEG-PNIPAM which was synthesized in our manuscript have passed FDA certification, except for PNIPAM. Molecular weight of PLGA, PEG and PNIPAM in our research was about 15000, 4000 and 2000 respectively. We have tried to reduce the proportion of PNIPAM in this polymer system. We are deeply sorry for our imprecise expression. Thanks a lot for the reviewer pointing it out. We have removed this sentence to make the discussion clearer in our manuscript.

Subsequently, the reviewer also mentioned “**DOX and ICG as building blocks for creating hybrid nano-capsules of 200 nm in diameter. All the experiments are well described and many of the results sounds, other like the EPR capacity of 200 nm nanocapsules is a fairy tale**”.

Generally speaking, nanoparticles (NPs) with a size of **100-200** nm can **improve** the circulatory half-life, but they are **not easy** to penetrate deep cellular layers near the tumor vessels (*Nature Nanotechnology*, 2011, 6, 385-391; *Chem. Soc. Rev.*, 2017, 46, 3830-3852; *Chem. Rev.* 2015, 115, 10410-10488; *Nat. Biotechnol.* 2015, 33, 941-951; *ACS Nano* 2015, 9, 7195-7206; *Nat. Biotechnol.* 2007, 25, 1165-1170.). On the contrary, **small** size NPs with a diameter of 4-20 nm **easily** penetrate into deep tumor tissues, but they are more prone to **rapid** clearance and **insufficient** drug

retention (*Nature nanotechnology* 2009, 4, 773-780). Therefore, it is **critically** desired to develop next-generation smart theranostic agents and nanocarriers (*ACS Nano* 2016, 10, 6753-6761; *Proc. Natl. Acad. Sci. USA* 2016, 113, 4164-4169) with a **large initial size** (100–200 nm) to cross biological barriers, to **provide** prolonged blood circulation and achieve high tumor accumulation, which then could be responsively **decomposed** into **small fragments** to release loaded guests and get **metabolized** promptly.

Hence, in our study, we synthesized nanocapsules with initial size of 200 nm to provide **long-lasting** blood circulation and less clearance of organs. But when the nanocapsules reach the tumor tissues, they could **shrink** and **decompose** into small-sized nanodrugs triggered by photothermal effect of ICG under laser irradiation and lower pH value in TME (Figure 4c-e). These capabilities show **significantly** enhanced intratumoral permeability to further improve the therapeutic effect in combination therapy with chemodynamic, photodynamic and chemotherapy (Figure 5, 6 and S27). In addition, we have tested real-time MRI of KB tumor-bearing mice without and without laser irradiation after intravenous injection of DOX-ICG@Fe/FeO-PPP-FA nanocapsules. After the irradiation of the laser, the MRI signal in the tumor area was darker (Supplementary Figure S27). These results confirm that laser-triggered shrinkage and decomposition of DOX-ICG@Fe/FeO-PPP nanocapsules were helpful for the deep tumor tissue penetration of nanocapsules. Following results were added to the revised manuscript:

Supplementary Figure S27. (a) Real-time MRI of KB tumor-bearing mice after intravenous injection of DOX-ICG@Fe/FeO-PPP-FA nanocapsules without and with the irradiation of laser (808 nm, 0.3 W cm⁻², 5 min) respectively. (b) The relative MRI signal intensities changing at the

tumor site respectively.

Next, the reviewer pointed out “Furthermore, a theoretical pH value of 5.4 within solid tumor is not possible, (this value is not compatible with the cell viability)”.

We have performed extensive additional experiments to provide all tests of these nanocapsules at pH=6.5 (Figure 3a, Figure 3c, Figure 4a, Figure 4b, Figure 4c, Supplementary Figure S10b and Supplementary Figure S22). These results also support our research.

The reviewer’s prospective “However, the main problem of this manuscript is that there are many other multifunctional systems that have been already published and many of them combine similar elements to produce a theragnostic system like the presented in this work. In addition, the presented manuscript isn’t a break-through in the topic, is a good manuscript that present nice results for being published elsewhere like ACS applied materials and interfaces or Nano-Research, but not in Nature Communication” is highly expected to change after considering above revisions and the fact that, our research system is significantly different from other reported systems. These differences include:

- (1) **Stimulatory** response of external laser and tumor microenvironment has prompted DOX-ICG@Fe/FeO-PPP nanocapsules to achieve synergistic effects of different cancer treatment modalities. Many other reported works of multiple stimulus responses **do not** focus on this aspect (*Angew. Chem. Int. Ed.* 2019, 58, 159-163; *Angew. Chem. Int. Ed.* 2018, 57, 17048-17052; *J. Am. Chem. Soc.* 2017, 139, 7522-7532; *J. Am. Chem. Soc.* 2014, 136, 14896-14902; *Mol. Pharmaceutics* 2014, 11, 1599-1610).
- (2) *In vitro* and *In vivo* experiments demonstrated that these nanocapsules can offer **remarkable** imaging and therapeutic results.
- (3) The materials that make up nanocapsules are basically FDA **certified** (PLGA, PEG and ICG). The furthermore experimental results show that the **biosafety** of the produced nanocapsules is relatively **reliable**.

(4) We have also explored the **metabolic problems** of the nanocapsules after cancer treatment, which is of great significance against the background of the great **challenge** of nano-biomedicine.

Following above changes and explanations, we strongly believe that the reviewer will reconsider to recommend our work for Nature Communications.

Response to Referee Three

Reviewer Comments:

This study reports a novel kind of near-infrared (NIR) light and tumor microenvironment (TME) dual responsive size-switchable nanocapsules (DOX-ICG@Fe/FeO-PPP). The smartly engineered DOX-ICG@Fe/FeO-PPP nanocapsules were shown to not only shrink and decompose into small-sized nanodrugs upon drug release, but also regulate TME to overproduce reactive oxygen species (ROS) for enhanced synergistic therapy in tumors. Sufficient experiments were performed to verify the magic theranostic performance of DOX-ICG@Fe/FeO-PPP nanocapsules. This paper is also well-written and well-organized. Therefore, I would like to recommend this paper for publication in Nature Communications after addressing the following issues:

Our response and revision:

We are thankful to the reviewer for positive comments and evaluation on our manuscript. We responded every comment from the referee one by one.

Q1. The mechanisms for the synthesis of the PLGA-PEG-PNIPAM (PPP) polymer and construction of DOX-ICG@Fe/FeO-PPP nanocapsules are suggested to be introduced in detail.

Our response and revision:

Thanks again to the reviewer for professional suggestions. The synthesis mechanisms were further elaborated and additional figures were added in revised manuscript as following:

“The formation of PPP was confirmed by the FTIR spectrometer, which was marked in Supplementary Figure S2. The red shift of the absorption peak for the stretching vibration of the C=O from carboxyl group (1635 cm^{-1}) to amide bond (1689 cm^{-1}) proves the amination of folic acid molecule (Supplementary Figure S2a: i and ii). The existence of vibration absorption peaks (3410 cm^{-1} and 1480 cm^{-1}) for N-H bond (Supplementary Figure S2a: iv) proved the successful synthesis of PLGA-FA. The formation of amide in PPP was also proved in Supplementary Figure S2b. Soon afterwards Fe/FeO NCs were tethered with PPP to form Fe/FeO-PPP heterostructures. Then, DOX and ICG were co-loaded into Fe/FeO-PPP heterostructures to develop flexible DOX-ICG@Fe/FeO-PPP nanocapsules by water-oil-water (W/O/W) emulsion method, which were used for subsequent experiments (*ACS Appl. Mater. Inter.* 2015, 7, 14896-14904; *J. Am. Chem. Soc.* 2013, 135, 835-843). We have introduced the mechanisms for the synthesis of the PLGA-PEG-PNIPAM (PPP). Water-oil ratio ($V_{\text{dichloromethane}}/V_{\text{water}}$) for the initial emulsion and PVA are two key conditions in the synthesis of nanocapsules. Water-oil ratio for the initial emulsion could affect the size and hollowness of the nanocapsule (Supplementary Figure S3a-e), and $V_{\text{dichloromethane}}/V_{\text{water}}=1:4$ was selected in the following experiments. The presence of PVA could affect the dispersion of nanocapsules (Supplementary Figure S3f). With the increase of the amount of Fe/FeO NCs, the number of nanocrystals on the shell also increased (Supplementary Figure S4). Finally, we chose to add 3 mg Fe/FeO NCs (Supplementary Figure S4b)”.

Supplementary Figure S3. TEM image of ICG@PPP nanocapsules in different condition: (a) $V_{\text{dichloromethane}}/V_{\text{water}}=1:1$, (b) $V_{\text{dichloromethane}}/V_{\text{water}}=1:2$, (c) $V_{\text{dichloromethane}}/V_{\text{water}}=1:4$, (d) $V_{\text{dichloromethane}}/V_{\text{water}}=1:6$, (e) $V_{\text{dichloromethane}}/V_{\text{water}}=1:8$ and (f) $V_{\text{dichloromethane}}/V_{\text{water}}=1:4$ and without PVA.

Supplementary Figure S4. TEM image of ICG@Fe/FeO-PPP nanocapsules in different concentration of Fe/FeO NCs: (a) $m_{\text{Fe/FeO NCs}}=1$ mg, (b) $m_{\text{Fe/FeO NCs}}=3$ mg, (c) $m_{\text{Fe/FeO NCs}}=6$ mg and (d) $m_{\text{Fe/FeO NCs}}=12$ mg.

Q2. The polydispersity index (PDI) values for the DLS results in Supplementary Figure S5 should be given.

Our response and revision:

We appreciate this valuable suggestion from reviewer. Polydispersity index (PDI) values were provided in revised Supplementary Figure S7. Following is the revised Figure:

Supplementary Figure S7. Hydrodynamic diameters measured by DLS for the ICG@PPP nanocapsules (203.8 ± 45.7 nm, PDI=0.224) (a) and DOX-ICG@Fe/FeO-PPP (218.9 ± 52.1 nm, PDI=0.238) nanocapsules dispersed in in PBS (b).

Q3. Where is the source of $^1\text{O}_2$? Why does the $^1\text{O}_2$ generation rely on pH? It seems that the Fenton reaction of ICG@Fe/FeO-PPP can only generate $\bullet\text{OH}$ rather than $^1\text{O}_2$.

Our response:

Thanks a lot to the reviewer's professional comment. There are two sources of $^1\text{O}_2$, one is the oxygen in tumor microenvironment, and the other is the air entering the cavity of nanocapsules during the synthesis of nanocapsules. The change in acidity of the system will affect the stability of nanocapsules, which will cause the amount of $^1\text{O}_2$ generation.

Q4. The generated $\bullet\text{OH}$ amount should be measured by using more accurate quantitative methods, such as methylene blue (MB) bleaching (*Angew. Chem. Int. Ed.* 2015, 54, 1770), terephalic acid (TA) oxidation (*Nano Lett.* 2017, 17, 4323).

Our response and revision:

We are thankful to the reviewer for this kind suggestion. We have utilized the terephalic acid (TA) oxidation to measure the generated $\bullet\text{OH}$. Revised discussions are included at Page 6 line 27-30, Page 7 line 1-2 and Page 18 line 1-9 and following are the revised Figures in our manuscript:

Figure 3. (b) Fluorescence intensity of TAOH at 440 nm as a function of laser irradiation time for the ICG@Fe/FeO-PPP nanocapsules and different control treatments. F_0 and F were the fluorescence intensities of the system without or with treatment, respectively. The error bars represent the standard deviations of three separate measurements.

Supplementary Figure S9. Detection of $\cdot\text{OH}$ generated by ICG@Fe/FeO-PPP nanocapsules.

Fluorescence spectra of TAOH induced by (a) only under 808 nm laser irradiation (0.3 W cm^{-2}), (b) ICG@PPP nanocapsules under 808 nm laser irradiation (0.3 W cm^{-2}), (c) Fe/FeO under 808 nm laser irradiation (0.3 W cm^{-2}) and (d) ICG@Fe/FeO-PPP nanocapsules under 808 nm laser irradiation (0.3 W cm^{-2}) from the same concentration of TA solution for different times (0–30 min). Inset in (b): After the oxidation of terephalic acid (TA) to 2-hydroxy- terephalic acid (TAOH) by $\bullet\text{OH}$, nonfluorescent TA was converted to fluorescent TAOH.

Q5. To validate the confocal fluorescence imaging result in Figure 3d, the intracellular ROS must also be measured through flow cytometry analysis.

Our response and revision:

Thanks very much for reviewer's suggestions. We have evaluated the $^1\text{O}_2$ generation by flow cytometry. The results of flow cytometry were consistent with those of confocal fluorescence imaging result in Figure 3d. The revised part was also highlighted in Page 7 line 17-19 and Page 18 line 24 in our manuscript. Followed is the revised figures in our manuscript:

Supplementary Figure S14. $^1\text{O}_2$ generation evaluated of control group by flow cytometry without DHR123 in KB cells under normoxic and hypoxic condition.

Supplementary Figure S15. $^1\text{O}_2$ generation evaluated of free ICG group by flow cytometry with DHR123 in KB cells under normoxic and hypoxic condition.

Supplementary Figure S16. $^1\text{O}_2$ generation evaluated of ICG@Fe/FeO-PPP nanocapsules group by flow cytometry with DHR123 in KB cells under normoxic and hypoxic condition.

Q6. The authors attribute the increased drug release of DOX-ICG@Fe/FeO-PPP nanocapsules to the instability of Fe/FeO NCs in weak acidic condition. Why was Fe/FeO NCs unstable in weak acidic condition? Can the authors provide convincing data?

Our response and revision:

Thanks a lot to reviewer for pointing towards this important aspect. Firstly, the activity of iron and iron oxides in nano-scale is much higher than that in macro-scale, and these could be ionized in weak acid environment (*Biomacromolecules* 2014, 15, 3171-3179; *Nat. Nanotech.*, 2016, 11, 977-985; *Nat. Nanotech.*, 2016, 11, 986-994).

The chemical reaction process is as follows:

Then we describe the contraction and degradation process of the nanocapsules *in vitro* in 48 hours in Figure 4c. Then, we extend the degradation time to 7 days under the same conditions, which further verifies our hypothesis (Supplementary Figure S22). The instability of Fe/FeO NCs in ICG@Fe/FeO-PPP nanocapsules under weak acid conditions was also proved. In addition, we observed the changes of ICG@Fe/FeO-PPP nanocapsules in KB cells by Bio-TEM (Supplementary Figure S23). These results further confirm the instability of Fe/FeO NCs under weak acid conditions. Following are the revised Figures in our manuscript:

Supplementary Figure S22. TEM image of the shrinking process for ICG@Fe/FeO-PPP nanocapsules after the irradiation of laser (808 nm, 0.3 W cm^{-2}) for 5 min (pH=6.5): (a) 0 d, (b) 2 d, (c) 7d.

Supplementary Figure S23. Bio-TEM images of KB cells incubated with ICG@Fe/FeO-PPP nanocapsules before (a) and after 4 h (b) under the irradiation of laser (808 nm, 0.3 W cm^{-2}) for 5 min.

Q7. There is no data to show the advantage of laser-triggered shrinkage of DOX-ICG@Fe/FeO-PPP nanocapsules in facilitating the deep tumor tissue penetration of nanocapsules.

Our response and revision:

We again thank the reviewer for pointing out another important aspect. Due to the spatial resolution of MRI, we tested real-time MRI of KB tumor-bearing mice

with and without laser irradiation after intravenous injection of DOX-ICG@Fe/FeO-PPP-FA nanocapsules. These results confirmed that the laser-triggered shrinkage of DOX-ICG@Fe/FeO-PPP nanocapsules was critical for the deep tumor tissue penetration of nanocapsules. Following results were added to the revised manuscript:

Supplementary Figure S27. (a) Real-time MRI of KB tumor-bearing mice after intravenous injection of DOX-ICG@Fe/FeO-PPP-FA nanocapsules without and with the irradiation of laser (808 nm, 0.3 W cm⁻², 5 min) respectively. (b) The relative MRI signal intensities changing at the tumor site respectively.

Q8. Some *in vitro* cell experiment results are suggested to be removed from supplementary information to article. Also, please add statistical analysis to Figure 4a, 4b, 5d, 6c, S16b, S20.

Our response and revision:

Thanks a lot to the reviewer for this kind suggestion. We removed the mentioned results from supplementary information and the corresponding statistical analysis results have been added to the Figure 4a, 4b, 5d, 6c, S16b and S20 in revised manuscript.

Q9. The quality of the figures should be improved. Currently the words in the figures are not shown clearly.

Our response and revision:

We are thankful to the reviewer for this kind suggestion. Wherever necessary, the

text font size was increased and we did our best to improve the resolution of figures according to the requirements of this journal. In addition, we have made it certain that the text in all figures of our manuscript is clearly readable. These corrections were highlighted in the revised manuscript.

REVIEWERS' COMMENTS:

Reviewer #1 (Remarks to the Author):

This paper reports the synthesis of near-infrared (NIR) light and tumor microenvironment (TME) dual responsive size-switchable nanocapsules (DOX-ICG@Fe/FeO-PPP), which not only shows dual responsive shrink and decomposition for drug release and tumor accumulation enhancement, but also can regulate TME to overproduce reactive oxygen species (ROS) for enhanced synergistic therapy in tumors. The prepared nanocapsules structure is relatively new and enlightening for controllable responsive drug delivery system design. Sufficient experiments were performed to verify the magic theranostic performance of DOX-ICG@Fe/FeO-PPP nanocapsules, which is of good scientific and novelty. All of the questions have been well addressed with sufficient experiments data. Therefore, I would like to recommend the revised paper for publication in Nature Communications.

Reviewer #2 (Remarks to the Author):

After reading the answers given by the authors, I still thinking that this manuscript is an incremental work of others that have been already published elsewhere.

Nevertheless and from a strictly scientific point I think that the most of the experiments are well done and the most of the conclusion sound and therefore this work could be published.

Reviewer #3 (Remarks to the Author):

The authors did a lot of work to address all the issues raised by the reviewers, so I'd like to recommend this nice paper for publication in Nat. Commun.